# Securing Multimodal AI through Internal Information Decomposition

**Jehyeok Yeon** [1]  **Hyeonjeong Ha** [1]  **Qiusi Zhan** [1]  **Heng Ji** [1]

## Abstract

Multimodal large language models introduce attack surfaces absent in unimodal systems: adversaries can distribute malicious intent across modalities to evade unimodal safeguards. This motivates using cross-modal consistency as a detection signal rather than inspecting each modality in isolation. Our key observation is that benign inputs induce compatible predictive behavior from text-only and vision-only reasoning that stabilizes when fused, whereas adversarial manipulation disrupts this consistency, causing abnormal multimodal behavior. Existing defenses that examine raw inputs or outputs overlook this internal fusion process, rendering them brittle and computationally expensive. We propose **FlowGuard**, a lightweight inference-time framework that detects harmful inputs by monitoring internal multimodal consistency. Unlike approaches that rely on scalar confidence metrics, FlowGuard derives *FlowVectors* inspired by Partial Information Decomposition that quantify cross-modal redundancy, synergy, and modality-specific dominance, capturing whether fused multimodal predictions remain aligned with unimodal semantic evidence. In a one-class classification problem trained solely on benign data, FlowGuard reduces Attack Success Rates from $> 90\%$ to $< 15\%$ on unseen attacks, with $< 3\%$ utility loss and up to a $6\times$ latency reduction. Our results demonstrate that monitoring cross-modal consistency offers an efficient and effective defense for multimodal reasoning.

## 1. Introduction

Multimodal large language models (MLLMs) are increasingly deployed in real-world systems such as autonomous navigation, assistive technologies, and medical support (Ji et al., 2025). Unlike unimodal models, MLLMs must integrate heterogeneous visual and textual signals during inference, introducing new safety risks that arise specifically from *multimodal fusion*. A defining characteristic of these failures is that harmful intent can be distributed across modalities: text-only and vision-only inputs may each appear benign and internally coherent, yet their combination induces unsafe behavior. Recent multimodal jailbreaks exploit this property by embedding malicious semantics into cross-modal interaction itself, allowing adversaries to bypass modality-specific, surface-level safeguards. Consequently, many safety failures in MLLMs are not detectable from raw pixels or tokens alone, but only emerge after the visual and textual information are combined during reasoning.

Existing multimodal safety defenses largely focus on sanitizing inputs (Jain et al., 2023), verifying generated outputs (Xu et al., 2024; Fares et al., 2024), or monitoring scalar confidence signals such as likelihood, entropy, or perplexity (Burns et al., 2024). While effective for identifying surface-level incoherence or low-confidence generations, these approaches do not examine how information from different modalities interacts inside the model, and often overfit to known attack patterns. In particular, they fail when text-only and vision-only reasoning each appear internally consistent, yet their fusion produces confident but compositionally misaligned behavior, where the multimodal prediction diverges from one or both unimodal predictions. This exposes a fundamental limitation of prior safety criteria, which emphasize fluency and confidence rather than the structure of multimodal integration.

In benign settings, visual and textual inputs typically induce compatible predictive behavior: unimodal predictions agree on shared semantics, contribute complementary evidence, and their fusion reduces uncertainty and stabilizes the model's decision. Under adversarial manipulation, this structure breaks down. The multimodal prediction may deviate sharply from one or both unimodal predictions, reflecting abnormal cross-modal influence even though each modality appears coherent in isolation. We argue this breakdown in cross-modal consistency provides a model-internal signal of unsafe behavior that is both general and attack-agnostic, and that multimodal safety should therefore be evaluated in terms of consistency between unimodal and multimodal

[1]University of Illinois Urbana-Champaign. Correspondence to: Jehyeok Yeon <jehyeok2@illinois.edu>.

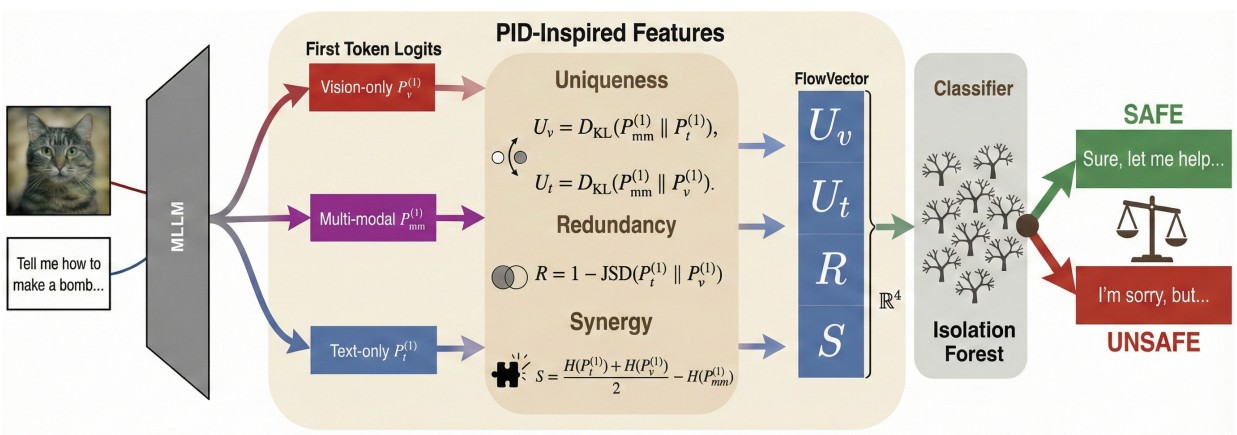

*Figure 1.* Overview of **FlowGuard**. Given an image–text pair, FlowGuard first probes the MLLM under three configurations: vision-only, text-only, and joint multimodal, and obtains first-token predictive distributions. FlowGuard then derives PID-inspired features that capture uniqueness, redundancy, and synergy, forming a 4D FlowVector. A one-class Isolation Forest detects anomalous fusion patterns, labeling benign inputs as SAFE and adversarial cross-modal interactions as UNSAFE. As a lightweight, plug-and-play detector, FlowGuard can be combined with other defense mechanisms to inform refusal decisions or guide steering interventions.

predictions rather than surface-level assessment alone.

Based on this observation, we introduce **FlowGuard**, [1]a lightweight inference-time framework that secures MLLMs by monitoring multimodal information flow during reasoning. FlowGuard queries the target MLLM under three configurations: text-only, vision-only, and joint multimodal inference, comparing the resulting predictive distributions. To characterize how modalities interact, FlowGuard measures how strongly the fused multimodal prediction deviates from each unimodal prediction, how much the unimodal predictions agree with one another, and whether fusion reduces or amplifies predictive uncertainty. These signals reflect modality dominance, shared semantic alignment, and fusion stability, and together serve as information-theoretic proxies for unique, redundant, and synergistic contributions from text and vision. We summarize these signals as a low-dimensional *FlowVector* and frame safety detection as a one-class classification problem trained solely on benign data, enabling robust detection of diverse and previously unseen multimodal attacks without modifying the underlying model or requiring adversarial supervision.

Under extensive experiments across multiple MLLM architectures and multimodal jailbreak benchmarks, FlowGuard consistently reduces Attack Success Rates from over $90\%$ to below $15\%$ on unseen attacks, while preserving benign utility within $3\%$ of baseline performance and operating up to $6\times$ faster than diffusion-based verification methods. These results demonstrate that monitoring cross-modal information consistency provides an efficient and robust alternative to existing surface-level safety mechanisms.

[1]Project page: `https://github.com/Jeybird248/FlowGuard`

**Contributions.** Our main contributions are:

- **Process-level multimodal safety.** We redefine MLLM safety in terms of consistency between text-only, vision-only, and joint predictive distributions, directly targeting multimodal fusion failures rather than surface-level detection.

- **FlowVectors for multimodal fusion.** We introduce compact information-theoretic features that capture redundancy, uniqueness, and synergy between modalities at inference time, characterizing model-internal information flow during multimodal reasoning.

- **Zero-shot multimodal attack detection.** We demonstrate that modeling benign multimodal behavior via one-class classification enables robust detection of diverse, unseen jailbreaks without adversarial training or generative verification.

## 2. Related Work

**Adversarial Attacks on Multimodal LLMs.** The integration of visual encoders with traditional large language models introduces vulnerabilities that arise specifically from multimodal fusion rather than from any single modality alone. Recent attacks exploit this property by distributing harmful intent across text and image channels such that neither modality appears unsafe in isolation, but their interaction induces policy-violating behavior. Text-based strategies such as Many-Shot Jailbreak (Ackerman & Panickssery, 2025), ArtPrompt (Jiang et al., 2024), and PAIR (Chao et al., 2024) manipulate linguistic context, formatting, or iteratively refined queries to bypass safety constraints. Vision-based attacks including FigStep (Gong et al., 2025)

and Visual Adversarial Jailbreak (VAJM) (Qi et al., 2023) embed harmful signals directly into the image channel via rasterized text or pixel-level optimization. Compositional cross-modal attacks such as Jailbreak in Pieces (Niu et al., 2024) and Universal Master Key (UMK) (Wang et al., 2024) jointly perturb both modalities, reliably inducing harmful behavior even when each modality appears benign in isolation.

**Defense Paradigms and Process-Level Signals.** Existing MLLM defenses primarily operate on surface-level representations, either by sanitizing inputs or verifying generated outputs. Previous works have established the utility of such cross-media consistency checking in the domain of misinformation detection (Fung et al., 2021). Generative verification methods such as MirrorCheck (Fares et al., 2024) and CIDER (Xu et al., 2024) rely on reconstruction, diffusion, or regeneration to assess semantic consistency, but incur substantial inference-time overhead and often degrade utility on benign but atypical inputs. Lightweight alternatives such as UniGuard (Oh et al., 2025) introduce external moderator models or prompt-level constraints, increasing architectural complexity while remaining sensitive to distribution shift.

More broadly, internal-signal approaches in unimodal LLMs explore logit-based indicators such as entropy, likelihood, or perplexity to flag anomalous behavior (Burns et al., 2024). However, multimodal jailbreaks frequently produce fluent, high-likelihood outputs that appear statistically normal when each modality is examined independently. As a result, scalar confidence or likelihood checks alone are insufficient for capturing failures that arise during multimodal fusion. Our work departs from prior defenses by targeting the *consistency between unimodal and multimodal predictive distributions* rather than the magnitude of any single signal. Rather than evaluating $P(y \mid x)$ alone, we contrast text-only, vision-only, and joint multimodal posteriors to detect systematic misalignment introduced during multimodal reasoning. This positions FlowGuard as complementary to input purification and output verification methods, providing a model-internal, inference-time signal that directly monitors failures in multimodal integration rather than surface-level representations.

## 3. Methodology

FlowGuard models multimodal safety as a property of consistency between unimodal and multimodal predictive behavior. Rather than treating safety as a property of surface inputs or generated outputs, we characterize whether visual and text reasoning integrate in a structurally normal way during inference.

In benign settings, visual and textual inputs contribute compatible evidence: they agree on shared semantic concepts while supplying distinct, modality-specific details that resolve mutual ambiguities. Under attack, this relationship is disrupted. Fusion no longer stabilizes prediction, and the multimodal posterior diverges from one or both unimodal pathways. FlowGuard detects such deviations by monitoring how information from text and images interacts in the model's output distributions against multimodal inputs.

### 3.1. Problem Formulation

We consider a multimodal large language model (MLLM) $\mathcal{M}$ that maps an image–text pair $(I, Q)$ to a generated response $Y$:

$$Y \sim P_{\mathcal{M}}(Y \mid I, Q),$$

where $I \in \mathcal{I} \subseteq \mathbb{R}^{H \times W \times C}$ denotes an image, $Q \in \mathcal{Q}$ a textual query, and $Y = (y_1, \ldots, y_T) \in \mathcal{Y}$ a token sequence over vocabulary $\mathcal{Y}$.

Let $\mathcal{Y}_{\mathrm{harm}} \subset \mathcal{Y}$ denote the set of unsafe outputs as defined by benchmark annotations and policy filters. A safety failure occurs when $Y \in \mathcal{Y}_{\mathrm{harm}}$ for some $(I, Q)$. Given an input $(I, Q)$ and the model $\mathcal{M}$, the goal is to determine whether the input is likely to induce unsafe behavior. The detector operates strictly at inference time and does not modify the underlying MLLM.

### 3.2. Modal Decomposition

To analyze the multimodal interaction structure, we probe the MLLM under three conditioning configurations—text-only, vision-only, and joint inference—and compare the resulting predictive distributions. In practice, we operate on the model's distribution at the first decoding step,

$$P^{(1)}(\cdot \mid I, Q) := P(y_1 \mid I, Q),$$

which captures the model's initial decision state prior to autoregressive amplification. Early decoding behavior strongly reflects alignment decisions such as refusal, compliance, or deflection. While later tokens evolve autoregressively, adversarial manipulation primarily alters the onset of generation, making $P^{(1)}$ a low-latency and informative statistic for detection (Qi et al., 2023; Zou et al., 2023). Section 5.5 empirically shows that, while FlowGuard can be extended to $k$ decoding steps, additional steps yield negligible gains relative to its computational cost.

We define two unimodal priors:

$$P_t^{(1)} = P_{\mathcal{M}}(\cdot \mid \emptyset, Q), \quad P_v^{(1)} = P_{\mathcal{M}}(\cdot \mid I, Q_{\emptyset}), \quad (1)$$

where $P_t^{(1)}$ captures the language prior induced by the query and $P_v^{(1)}$ isolates visual semantics using a fixed neutral prompt $Q_{\emptyset}$ (e.g., "Describe this image"). Appendix B.7 verifies robustness to the choice of a specific neutral prompt.

These are compared against the fused multimodal posterior:

$$P_{mm}^{(1)} = P_{\mathcal{M}}(\cdot \mid I, Q). \tag{2}$$

Under aligned reasoning, unimodal priors induce compatible predictive structure, and the fused posterior integrates them to reduce predictive uncertainty relative to the unimodal baselines. Adversarial inputs disrupt this balance, producing abnormal relationships where the $P_{mm}^{(1)}$ departs sharply from one or both unimodal reasoning pathways. Measuring how $P_{mm}^{(1)}$ relates to $P_t^{(1)}$ and $P_v^{(1)}$, FlowGuard exposes failures in multimodal interaction that are not visible from any single modality alone.

### 3.3. PID Motivation for Multimodal Interaction

FlowGuard is inspired by Partial Information Decomposition (PID) (Williams & Beer, 2010), which characterizes how multiple sources jointly inform a target via redundancy, uniqueness, and synergy. Given sources $X_1, X_2$ and target $Y$, PID decomposes $I(Y; X_1, X_2)$ into information shared by both sources, information unique to each, and information that arises only from their joint interaction. In the MLLM setting, we hypothesize that the unimodal reasoning pathways act as sources and the fused multimodal prediction acts as the target. Redundancy reflects agreement between text and vision, uniqueness reflects modality dominance, and synergy reflects whether fusion reduces uncertainty beyond either modality alone.

PID requires access to full joint distributions over high-dimensional variables, which makes finding the exact values computationally intractable for modern MLLM posteriors. FlowGuard therefore does not compute PID explicitly but instead adopts its conceptual structure: multimodal safety is characterized by how information from text and vision overlaps, diverges, and stabilizes under fusion. We extract low-dimensional proxies that reflect these relationships directly from the model's output distributions at inference time.

### 3.4. FlowVectors: Information-Theoretic Features

Each input $x = (I, Q)$ is mapped to a FlowVector $\phi(x) = (U_v, U_t, R, S) \in \mathbb{R}^4$, summarizing how information from the unimodal priors interacts in the fused multimodal posterior. These features instantiate PID-inspired notions using entropy and divergence over token distributions. Here $H(\cdot)$ denotes Shannon entropy (Shannon, 1948) over the token probability distribution, $D_{\mathrm{KL}}$ denotes Kullback–Leibler divergence (Kullback & Leibler, 1951), and JSD($\cdot \parallel \cdot$) denotes the Jensen–Shannon divergence (Lin, 1991).

**Uniqueness.** We measure modality dominance via directional divergence:

$$U_v = D_{\mathrm{KL}}(P_{mm}^{(1)} \parallel P_t^{(1)}), \quad U_t = D_{\mathrm{KL}}(P_{mm}^{(1)} \parallel P_v^{(1)}). \tag{3}$$

These quantities capture how strongly the fused posterior diverges from each unimodal prior. In aligned behavior, fusion preserves a balanced influence between modalities. Adversarial manipulation breaks this balance: text-only jailbreaks suppress visual grounding, while visual injections override linguistic priors, producing asymmetric divergence signals captured by $U_v$ and $U_t$.

**Redundancy.** We estimate agreement between unimodal priors as

$$R = 1 - \mathrm{JSD}(P_t^{(1)} \parallel P_v^{(1)}). \tag{4}$$

where the JSD is computed in bits (base 2) and thus bounded to $[0, 1]$. This captures semantic alignment between text-only and vision-only predictions. Benign inputs induce aligned compatible structure, while adversarial prompts introduce conflicting intent, reducing redundancy.

**Synergy.** We approximate fusion stability via entropy reduction:

$$S = \frac{H(P_t^{(1)}) + H(P_v^{(1)})}{2} - H(P_{mm}^{(1)}). \tag{5}$$

This measures whether fusion reduces predictive uncertainty beyond either alone. Constructive integration typically yields $S > 0$. Under adversarial constraints, the fused posterior may become more entropic than its components ($S < 0$), indicating unstable interaction. FlowGuard uses $S$ jointly with redundancy and uniqueness to mitigate false positives on ambiguous but benign inputs.

Each KL term in $U_v$ and $U_t$ asks how costly it is to explain the fused posterior using one unimodal prior alone. The directionality $D_{\mathrm{KL}}(P_{mm}^{(1)} \parallel P_t^{(1)})$ becomes large when fusion assigns mass to tokens unlikely under text-only reasoning, capturing modality-specific influence introduced by vision; the reverse direction captures the symmetric case for text. Redundancy uses Jensen–Shannon divergence because it is symmetric, bounded in $[0, 1]$ when computed in bits, and well-suited to comparing agreement between two unimodal priors prior to fusion. Synergy treats the average unimodal entropy as an uncertainty baseline and asks whether multimodal fusion resolves or amplifies it: positive $S$ reflects constructive fusion, while negative $S$ indicates that fusion destabilizes the prediction. Each feature is ambiguous on its own. For example, high redundancy can occur in both benign and cooperative-attack regimes, uniqueness only diagnoses asymmetric dominance, and synergy detects instability without identifying the source. FlowGuard combines

all four into a single 4D vector that resolves these ambiguities jointly.

## 3.5. Defense Objective

We formulate safety detection as one-class classification over benign multimodal information flow patterns: aligned inputs occupy a compact region in FlowVector space, while adversarial inputs appear as outliers due to abnormal cross-modal interactions. We train an Isolation Forest (Liu et al., 2008) on benign FlowVectors, yielding attack-agnostic detection without attack supervision or modification of the underlying MLLM.

This choice reflects the nonstationarity of multimodal jailbreaks: any fixed set of labeled attacks under-covers the deployment distribution, so a supervised classifier risks overfitting to specific perturbation patterns. Statistically, FlowVectors form a compact benign cluster in $\mathbb{R}^4$, whereas raw logits exhibit high-dimensional variation that masks the cross-modal signal. Empirically, a supervised MLP attains high in-distribution AUC on its training attack but collapses on unseen ones, whereas Isolation Forest on FlowVectors maintains AUC $\gtrsim 0.88$ across out-of-distribution settings (Appendix B.5).

**Access requirement.** FlowGuard does not require model weights, gradients, or hidden activations. It requires only access to the next-token probability distribution, or to a sufficiently large top-$k$ logprob tail when the target is queried through an API that provides this information. This places FlowGuard between full white-box access and pure black-box text-only APIs; we validate the partial-distribution setting on GPT-4.1-mini in Section 4.2 and Appendix D.

## 4. Experimental Setup

### 4.1. Training and Implementation Details

**One-Class Classification.** FlowGuard models the distribution of aligned multimodal information flow using only benign data. We sample $N = 10{,}000$ image–question pairs from the VQAv2 validation split (Goyal et al., 2017), disjoint from other safety benchmarks. This dataset serves as a proxy for aligned behavior. The sampling seed is fixed across runs unless otherwise stated. Appendix B.8 shows that performance saturates at approximately $N \approx 2{,}000$ samples, indicating low sample complexity.

**Detector Configuration.** We implement the anomaly detector using the standard Isolation Forest algorithm, training per model. We use Isolation Forest with default thresholding (`contamination='auto'`, decision boundary at normalized score 0.5), trained per target model. Sensitivity to contamination and comparisons against Autoencoders and One-Class SVMs are in Appendices B.5 and B.3.

### 4.2. Target Models

We evaluate FlowGuard on three open-weight vision-language model families: **LLaVA-1.5-7B** (Liu et al., 2023), **Qwen2.5-VL-7B-Instruct** (Bai et al., 2025), and **Gemma-3-4B** (Kamath et al., 2025), covering CLIP-initialized and unified multimodal transformers. To assess scaling behavior, we additionally evaluate a **LLaMA-3.1-70B** (Dubey et al., 2024) multimodal variant, instantiated as the `LLaMA-3.1-70B-Instruct` language backbone paired with a CLIP ViT-L/14-336 vision encoder. To evaluate the partial-distribution access setting, we further evaluate **GPT-4.1-mini** (OpenAI, 2025), where only top-$k$ logprobs are exposed via the API; FlowVectors are approximated from the truncated tail. FlowGuard requires only next-token predictive distributions (or top-$k$ logprobs for API models); no weights, gradients, or activations are accessed.

### 4.3. Evaluation Benchmarks

We evaluate performance along two complementary axes:

**Safety Assessment (Unsafe Inputs).** This axis measures detection performance on inputs containing harmful content or adversarial manipulation. We utilize **MM-SafetyBench** (Liu et al., 2024) and **VLSafe** (Chen et al., 2024) for standard policy violations (e.g., hate speech and physical harm), and the unsafe subsets of **VLSU** (Palaskar et al., 2025) to isolate compositional cross-modal threats where malicious content is distributed across modalities.

**Utility Assessment (Safe Inputs).** This axis measures false positive rates on benign inputs to ensure the defense is not oversensitive. We use **MOSSBench** (Li et al., 2024), which consists of benign but semantically nuanced queries that frequently trigger false refusals, alongside **VizWiz-VQA** (Gurari et al., 2018) and the safe subsets of **VLSU**. These datasets stress-test FlowGuard under atypical but non-adversarial inputs.

### 4.4. Threat Configuration

We evaluate FlowGuard under unsafe inputs that include both explicit harmful queries and multimodal jailbreak attacks. These attacks instantiate the modality-specific threat categories described below and cover obfuscation-based, optimization-based, and compositional strategies. All optimization-based attacks are executed with standardized budgets and iteration counts across models (Table 4 in Appendix A.2.6), to ensure fair cross-model comparison.

**Text-Only Attacks.** We evaluate **Many-Shot Jailbreak (MSJ)** (Ackerman & Panickssery, 2025), which exploits long-context in-context learning to override safety training; **ArtPrompt** (Jiang et al., 2024), which uses ASCII-style formatting to obfuscate sensitive trigger words from

text-based filters; and **PAIR** (Chao et al., 2024), which iteratively refines text-only prompts against the target model in a black-box manner.

**Image-Only Attacks.** We include **FigStep** (Gong et al., 2025), a typographic attack that embeds forbidden instructions as rasterized text; **Visual Adversarial Jailbreak (VAJM)** (Qi et al., 2023), which applies projected gradient descent to introduce imperceptible pixel-level perturbations; and **APGD** (Croce & Hein, 2020), a PGD-based image-only attack that maximizes harmful affirmative-response likelihood through pixel-level perturbations.

**Cross-Modal Attacks.** We evaluate **Universal Master Key (UMK)** (Wang et al., 2024), an optimization-based strategy that jointly perturbs an image prefix and a text suffix to bypass safety alignment; **Jailbreak in Pieces** (Niu et al., 2024), a compositional attack that pairs an embedding-targeted adversarial image with a generic textual prompt so that harmful semantics emerge only after fusion; and **Color-Based Substitution Cipher (CBSC)** (Niu et al., 2024), which encodes forbidden keywords via visual color cues so that the harmful instruction is recoverable only when the image and text are interpreted jointly.

### 4.5. Baselines

We compare FlowGuard against representative inference-time defenses spanning purification, verification, and prompt-based alignment. These include **CIDER** (Xu et al., 2024), which detects attacks via semantic shifts between original and diffusion-denoised images; **MirrorCheck** (Fares et al., 2024), which applies cycle-consistency via auxiliary reconstruction; **UniGuard-P** (Oh et al., 2025), a prompt-based alignment strategy; and **Llama Guard 4 12B** (Inan et al., 2023), a state-of-the-art multimodal safety classifier that filters unsafe content by evaluating inputs against a predefined hazard taxonomy. We also compare against **ECSO** (Gou et al., 2024), which removes visual input post hoc to detect visual injections.

Additionally, to isolate the specific contribution of the information-theoretic structure from generic latent irregularities, we evaluate a **Raw Embedding (Raw Emb.)** baseline. This control applies the same anomaly detection protocol used in FlowGuard on the MLLM's high-dimensional final-layer hidden states, testing whether adversarial inputs can be detected as simple geometric outliers, or if they specifically require the *relational decomposition* captured by our framework. To ensure strict comparability, all baselines are executed using their authors' recommended configurations without task-specific tuning, preserving identical deployment assumptions.

### 4.6. Evaluation Protocol

For each benchmark and attack configuration, we report the *Attack Success Rate (ASR)*, defined as the fraction of prompts that result in a harmful model response. Harmfulness is determined using benchmark annotations and policy categories, supplemented with automatic rule-based filters when necessary. To validate the automated metrics, a random subset of $N = 100$ responses was manually verified, observing a 96% agreement rate between the automated results and human judgments. FlowGuard operates as a pre-generation detector. When an input is flagged as anomalous, generation is suppressed and the attempt is counted as safe. Otherwise, the model generates normally and the output is evaluated for harmfulness. A lower ASR therefore directly reflects detection effectiveness.

We additionally report *AUROC* as a threshold-free measure of separation between benign and adversarial FlowVectors, computed by sweeping the Isolation Forest anomaly score across pooled (benign, adversarial) samples. ASR captures deployment behavior at the operating threshold, whereas AUROC captures the underlying detector quality independent of threshold choice. Per-attack FlowGuard AUROC values are reported in Appendix B.6. We further report the *False Positive Rate (FPR)*, defined as the fraction of benign inputs flagged as anomalous, to quantify over-refusal under benign distribution shift.

## 5. Results

We evaluate FlowGuard along three axes: (i) effectiveness against unsafe inputs and adversarial jailbreaks, (ii) preservation of vision-language utility on benign data, and (iii) computational efficiency and robustness of the detection signal.

### 5.1. Effectiveness Against Unsafe Inputs and Jailbreaks

Table 1 reports the Attack Success Rate (ASR) on LLaVA-1.5-7B, with results from additional MLLMs provided in Appendix D. We report the mean ASR and standard deviation across five independent runs with different random seeds to assess stability. Without defense, models exhibit high vulnerability to multimodal attacks, with ASR exceeding 90% on advanced methods like Universal Master Key (UMK) and Color-Based Substitution Cipher (CBSC). Existing defenses demonstrate modality-specific behavior but limited generalization. For example, ECSO substantially reduces visual attack success (e.g., $\approx 10\%$ on FigStep and VAJM), yet remains ineffective against text-dominant threats. CIDER and MirrorCheck similarly struggle with compositional attacks such as *Jailbreak in Pieces*, allowing ASR above $25\%$ on VLSU. Llama Guard 4 also shows vulnerability to complex cross-modal obfuscation, performing exceptionally well on unimodal attacks while struggling on substitution ciphers,

*Table 1.* **Comparative Attack Success Rate (ASR) on LLaVA-1.5-7B.** Mean $\pm$ Std. Dev. over 5 runs. FlowGuard achieves the strongest overall cross-modal defense profile, consistently reducing ASR across text, visual, and cross-modal attacks while preserving utility. The best results are **bolded** and the second-best are underlined. Lower is better.

| Data | Type | Attack Method | Base | CIDER | Mirror. | UniGuard | Llama Guard 4 | ECSO | Raw Emb. | FlowGuard |
|---|---|---|---|---|---|---|---|---|---|---|
| MMSB | - | Direct Queries | 38.2±1.1 | 32.6±1.3 | 31.9±1.2 | 14.8±1.9 | 6.2±0.5 | 10.4±1.1 | 24.5±2.4 | **3.2±0.4** |
| | Text | MSJ | 58.4±0.7 | 53.9±0.9 | 54.2±1.0 | 44.8±1.9 | **8.5±0.9** | 57.1±0.8 | 41.6±2.3 | 14.2±0.6 |
| | | ArtPrompt | 52.0±1.2 | 45.3±1.6 | 44.9±1.5 | 39.5±2.1 | **5.8±0.8** | 49.2±1.4 | 36.8±3.8 | 9.1±0.5 |
| | | PAIR | 53.4±1.0 | 47.8±1.4 | 47.4±1.4 | 41.0±2.0 | **6.5±0.8** | 51.6±1.3 | 38.2±3.5 | 11.8±0.6 |
| | Visual | FigStep | 84.0±0.8 | 15.6±1.1 | 17.2±1.3 | 68.1±1.9 | **4.1±1.1** | 9.8±0.8 | 39.4±3.4 | 7.2±0.5 |
| | | VAJM | 51.0±1.5 | 11.2±0.7 | 12.5±0.8 | 44.3±2.0 | 11.5±1.0 | 6.7±0.5 | 32.5±2.7 | **6.1±0.5** |
| | | APGD | 56.2±1.3 | 12.8±0.8 | 14.1±0.9 | 47.5±2.0 | 9.8±1.0 | 7.9±0.6 | 35.2±2.9 | **6.4±0.5** |
| | Cross | UMK | 96.0±0.4 | 21.8±1.2 | 23.6±1.3 | 82.4±1.7 | 11.5±1.5 | 12.4±0.9 | 52.8±3.6 | **8.7±0.6** |
| | | CBSC | 94.8±0.5 | 23.9±1.3 | 25.4±1.4 | 80.7±1.8 | 20.2±1.4 | 13.9±1.0 | 51.6±3.4 | **9.1±0.6** |
| | | Jailbreak in Pieces | 87.0±1.0 | 23.4±1.4 | 25.1±1.5 | 70.8±2.1 | 8.8±1.2 | 14.3±1.0 | 42.2±3.1 | **8.4±0.6** |
| VLSafe | - | Direct Queries | 39.6±1.0 | 33.9±1.2 | 33.2±1.3 | 16.2±1.8 | 5.9±0.4 | 11.5±1.0 | 26.4±2.5 | **3.4±0.4** |
| | Text | MSJ | 56.9±0.8 | 52.4±1.0 | 52.8±1.1 | 43.5±2.0 | **7.2±0.8** | 55.7±0.8 | 40.2±2.2 | 13.6±0.6 |
| | | ArtPrompt | 55.4±1.3 | 48.7±1.6 | 48.1±1.5 | 42.6±2.0 | **4.5±0.7** | 52.0±1.4 | 38.5±3.7 | 9.7±0.5 |
| | | PAIR | 54.2±1.1 | 48.6±1.4 | 48.0±1.4 | 41.8±2.0 | **6.0±0.8** | 52.5±1.3 | 39.0±3.5 | 11.2±0.6 |
| | Visual | FigStep | 81.6±1.1 | 14.2±1.0 | 15.9±1.2 | 66.7±1.8 | **6.4±1.1** | 9.1±0.7 | 37.8±3.0 | 6.9±0.5 |
| | | VAJM | 53.2±1.4 | 12.0±0.8 | 13.4±0.9 | 45.9±2.1 | 10.8±1.0 | 7.3±0.6 | 33.9±2.9 | **6.3±0.5** |
| | | APGD | 58.0±1.3 | 13.5±0.9 | 14.7±1.0 | 49.2±2.1 | 10.2±1.0 | 8.4±0.6 | 36.1±3.0 | **6.7±0.5** |
| | Cross | UMK | 95.1±0.5 | 23.4±1.2 | 24.7±1.3 | 83.6±1.6 | 9.8±1.4 | 13.7±0.9 | 54.0±3.5 | **8.3±0.6** |
| | | CBSC | 93.5±0.6 | 24.9±1.3 | 26.1±1.4 | 82.0±1.7 | 19.1±1.3 | 15.1±1.0 | 52.7±3.4 | **8.9±0.6** |
| | | Jailbreak in Pieces | 89.4±0.9 | 25.1±1.5 | 26.8±1.6 | 72.3±2.0 | 11.9±1.2 | 15.8±1.1 | 44.2±3.2 | **8.8±0.6** |
| VLSU | - | Direct Queries | 36.8±1.1 | 31.2±1.3 | 30.6±1.2 | 13.6±1.7 | 5.5±0.4 | 9.8±1.0 | 23.7±2.3 | **3.1±0.4** |
| | Text | MSJ | 54.7±0.9 | 50.6±1.1 | 51.0±1.2 | 42.1±2.1 | 13.9±0.8 | 53.4±0.9 | 38.7±2.4 | **12.9±0.6** |
| | | ArtPrompt | 48.2±1.5 | 42.1±1.7 | 41.7±1.6 | 36.9±2.1 | 13.2±0.7 | 45.6±1.5 | 33.4±3.8 | **8.6±0.5** |
| | | PAIR | 50.1±1.2 | 44.0±1.5 | 43.6±1.5 | 38.2±2.1 | 13.5±0.7 | 47.8±1.4 | 34.2±3.6 | **10.6±0.6** |
| | Visual | FigStep | 85.3±1.0 | 17.8±1.2 | 19.2±1.3 | 69.5±1.9 | 22.7±1.1 | 11.2±0.9 | 40.3±3.2 | **7.8±0.6** |
| | | VAJM | 50.5±1.6 | 13.6±0.9 | 14.9±1.0 | 43.8±2.0 | 19.5±1.0 | 8.4±0.7 | 32.7±2.9 | **6.7±0.5** |
| | | APGD | 55.8±1.4 | 14.6±0.9 | 15.9±1.0 | 46.8±2.0 | 18.5±1.1 | 9.1±0.7 | 34.8±3.0 | **7.0±0.5** |
| | Cross | UMK | 93.8±0.6 | 24.6±1.3 | 26.1±1.4 | 81.9±1.7 | 28.4±1.3 | 14.6±1.0 | 52.9±3.4 | **8.4±0.6** |
| | | CBSC | 92.5±0.7 | 26.1±1.4 | 27.4±1.5 | 80.4±1.8 | 17.5±1.3 | 15.8±1.1 | 51.5±3.3 | **8.8±0.6** |
| | | Jailbreak in Pieces | 84.2±1.1 | 26.4±1.5 | 27.8±1.6 | 71.2±2.1 | 25.1±1.2 | 16.9±1.1 | 41.5±3.1 | **8.9±0.6** |

with ASR rising to 20.2% on CBSC (MMSB).

FlowGuard maintains a more consistent defense profile across all modalities, achieving ASR $\leq 14.2\%$ across all evaluated attacks and an overall FlowGuard AUROC of 0.942 (per-attack AUROC in Appendix B.6). It outperforms Llama Guard 4 on complex cross-modal vectors (e.g., reducing UMK ASR to 8.7% vs. 11.5% and CBSC to 9.1% vs. 20.2%). Similar results hold on **Qwen2.5-VL** and **Gemma-3**, on the substantially larger **LLaMA-3.1-70B** (7.9% average ASR), and under partial-distribution access on **GPT-4.1-mini** where only top-$k$ logprobs are exposed via the API (10.4% average ASR; Appendix D). The FlowVector signal therefore persists across the 4B–70B open-weight range and recovers from truncated logprob tails, rather than depending on a specific architecture or full-distribution access. The low variance across runs (generally $< \pm 0.6\%$) suggests that monitoring multimodal information flow yields consistent decisions compared to stochastic purification or prompt-based strategies.

## 5.2. Preservation of Vision-Language Utility

A safety mechanism must avoid degrading performance on benign inputs. We measure utility as task accuracy relative to the undefended model on three benchmarks: **VQAv2** (standard in-distribution using the held-out validation set), **VizWiz-VQA** (out-of-distribution, low-quality images), and **MOSSBench** (benign but semantically complex and sensitive queries). As shown in Table 2, FlowGuard achieves the most favorable safety-utility balance, maintaining accuracy within **2.4%** of the unprotected baseline. In contrast, generative verification methods incur larger drops under distribution shift; CIDER and MirrorCheck degrade substantially on VizWiz (to $\approx$ 35–38% accuracy) due to reconstruction artifacts on low-quality images.

Prompt-based and classifier-based defenses, while robust on standard visual tasks, struggle with semantic subtlety. **Uni-Guard** (65.4%) and **Llama Guard 4** (74.8%) both show notable performance drops on MOSSBench. This reflects

a tendency toward over-defensiveness, where safe queries containing sensitive language trigger false refusals. By operating on internal consistency rather than surface-level content filtering or regeneration, FlowGuard avoids these failure modes, maintaining stable accuracy across both in-distribution and out-of-distribution benign inputs.

Quantitatively, FlowGuard yields a $2.4\%$ FPR under the default threshold, compared with $11.2\%$ for perplexity-based and $14.5\%$ for confidence-based detection (Appendix B.4), and maintains $79.5\%$ accuracy on MOSS-Bench, the strongest among evaluated defenses on the over-refusal axis.

### 5.3. Efficiency and Sensitivity Analysis

**Computational Efficiency.** Figure 2 compares detection performance against inference latency for FlowGuard and competing defenses. Latency is measured end-to-end on identical hardware, including all defense-specific computation (e.g., three forward passes for text-only, vision-only, and multimodal inference) but excluding base model generation. FlowGuard achieves an average F1-score of 0.94 with a latency of approximately 1.3 seconds per sample. While UniGuard offers a marginal speed advantage (1.1s), it suffers a significant performance drop (0.72 F1). Conversely, Llama Guard 4 provides competitive robustness (0.88 F1) but incurs nearly double the latency (2.3s). FlowGuard balances efficiency and safety. In contrast, diffusion-based methods such as MirrorCheck and CIDER incur substantially higher computational costs (4.5s and 8.5s, respectively) without achieving comparable detection performance.

### 5.4. Component Analysis & Feature Importance

We analyze the contribution of each FlowVector component on both benign (VQAv2, MOSSBench, VizWiz) and adversarial (MMSB, VLSafe, VLSU) benchmarks.

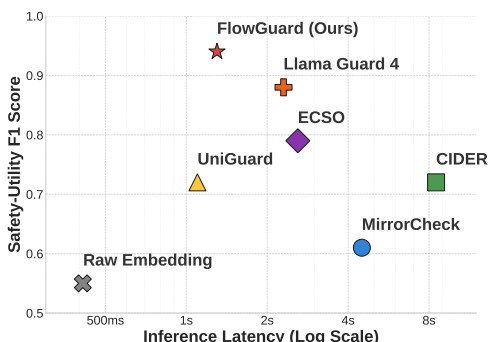

*Figure 2.* **Latency vs. Detection Performance.** FlowGuard achieves a favorable trade-off between detection F1-score and inference latency compared to existing defenses.

**Role of Synergy ($S$).** Figure 3 presents the results of a leave-one-feature-out ablation. Excluding Synergy ($S$) produces the largest performance decrease, with AUC drops of **-0.125** on VLSafe and **-0.110** on VLSU. This indicates that deviations in multimodal uncertainty reduction are a primary signal for detecting adversarial interaction. Conversely, **Visual Uniqueness** ($U_v$) appears to play a complementary role, preventing a **-0.150** drop on VizWiz, suggesting that visual dominance signals help stabilize detection on low-quality or atypical images.

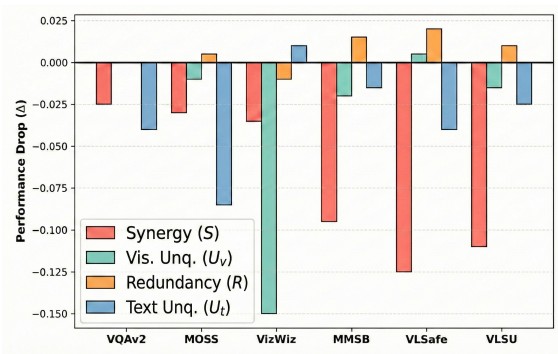

*Figure 3.* **Feature Necessity (Ablation).** Negative values indicate the feature is essential. The exclusion of **Synergy** ($S$) results in the largest performance drops on attack datasets (Right), while **Visual Uniqueness** ($U_v$) is critical specifically for OOD VizWiz data.

**Standalone Predictive Power.** Figure 5 evaluates each feature independently. **Synergy** ($S$) achieves standalone AUC scores exceeding **0.88** across attack benchmarks, while **Redundancy** ($R$) performs significantly worse on complex attacks (e.g., VLSU AUC $\approx 0.32$). This suggests that jailbreaks may preserve surface-level semantic agreement while actually disrupting deeper behavior captured by $S$.

**Embedding visualization.** Figure 4 compares t-SNE projections of raw first-step logit embeddings against 4D FlowVectors for the same set of benign and adversarial samples. Raw logits exhibit substantial overlap between the two classes, whereas FlowVectors form a compact benign cluster with adversarial points scattered along its periphery. This separation supports the use of relational cross-modal features rather than direct logit-space anomaly detection.

### 5.5. Ablation on Number of Decoding Steps

We explore the latency–quality trade-off by varying the number of decoding steps $k$ monitored by FlowGuard (Table 3). Moving from $k = 1$ to $k = 10$ improves AUC by at most $+0.006$ while latency scales linearly, so the detection signal is already saturated at the first decoding step and $k = 1$ is the efficient operating point. However, $2.1\%$ of adversarial samples that evade $k = 1$ are caught at $k = 3$, providing a practically meaningful safety margin against attacks that delay harmful semantics to later tokens. Adaptive temporal probing, in which $k > 1$ is invoked only when the latency budget permits or the $k = 1$ anomaly score lies near the deci-

*Table 2.* **Utility Preservation on Benign Benchmarks (Effective Accuracy).** We report the accuracy (Mean ± Std. Dev. over 5 seeds) of the target models when protected by different defenses. **Base** represents the raw model performance (upper bound). FlowGuard incurs the smallest performance drop with the highest stability. The best score is **bolded** and the second best is underlined.

| Benchmark | Base | CIDER | Mirror. | UniGuard | Llama Guard 4 | ECSO | Raw Emb. | FlowGuard |
|---|---|---|---|---|---|---|---|---|
| **VQAv2** | 79.4±0.4 | 74.5±1.1 | 75.1±1.2 | 78.2±0.6 | **78.9**±0.5 | 76.5±0.5 | 68.4±1.8 | 77.1±0.3 |
| **VizWiz-VQA** | 56.2±0.5 | 35.2±2.1 | 38.1±1.9 | **55.4±0.8** | 55.1±0.6 | 51.2±0.7 | 41.5±2.4 | 53.8±0.4 |
| **MOSSBench** | 82.1±0.6 | 76.8±1.5 | 77.5±1.4 | 65.4±2.2 | 74.8±1.2 | 78.1±0.9 | 58.2±2.5 | **79.5**±0.5 |
| **Avg. Drop ($\Delta \downarrow$)** | - | 10.4 | 9.0 | 6.2 | 3.0 | 4.0 | 16.5 | **2.4** |

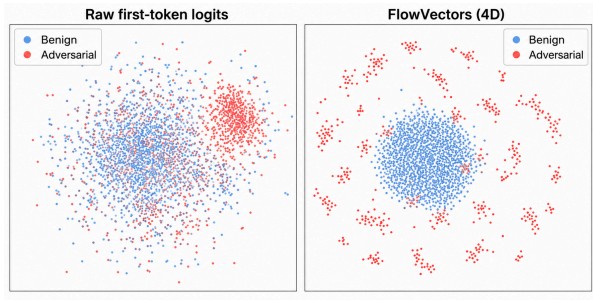

*Figure 4.* **t-SNE visualization of benign and adversarial samples using raw first-step logits versus FlowVectors.** Raw logits exhibit substantial overlap between inputs, while FlowVectors produce clearer separation, supporting the need for relational cross-modal features rather than direct logit-space anomaly detection.

sion boundary, retains the efficiency of single-step detection while offering a built-in defense against future attacks engineered to suppress first-token signals and surface adversarial behavior only after several decoding steps.

*Table 3.* **Decoding step efficiency.** Deeper monitoring yields marginal AUC gains at linear latency cost.

| Steps ($k$) | AUC | Latency (s) | $\Delta$ AUC vs $k = 1$ |
|---|---|---|---|
| **1** | **0.942** | **1.3** | – |
| 3 | 0.944 | 3.9 | +0.002 |
| 5 | 0.946 | 6.5 | +0.004 |
| 10 | 0.948 | 12.8 | +0.006 |

## 5.6. Adversarial Robustness Analysis

We probe two adaptive settings designed to defeat Flow-Guard directly. A white-box adversary with gradient access augments the jailbreak loss with a stealth penalty pulling the FlowVector toward the benign mean (Appendix B.1); across stealth weights $\lambda$, no setting simultaneously achieves high ASR and a benign anomaly score, and AUROC remains $\geq 0.78$ until the constraint forces the attack to collapse outright. A second stress test co-optimizes GCG and PGD toward the target completion "Sure, here is" so both modalities push the same top token (Appendix B.2); undefended ASR rises to 86.4%, but FlowGuard reduces it to 11.2%, with synergy collapsing from $S = +0.42$ on benign inputs to $S = -0.31$ under attack. Token-level agreement does not propagate to the rest of the posterior, and FlowGuard's KL-, JSD-, and entropy-based features compare full distri-

butions rather than argmax tokens. Both findings reflect a structural tension between harmful target induction and benign FlowVector geometry: attackers can satisfy stealth or success, but not both at once.

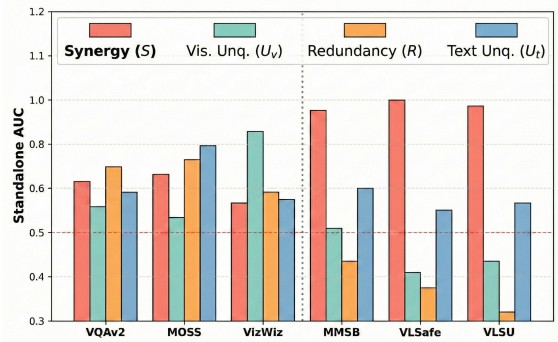

*Figure 5.* **Feature Predictive Power (Standalone AUC). Synergy** ($S$) consistently achieves the highest separation on adversarial benchmarks ($> 0.88$). In contrast, **Redundancy** ($R$) performs well on benign data but provides limited discriminative power on compositional attacks (VLSU).

## 6. Conclusion and Future Work

We proposed shifting MLLM safety from surface-level inspection to the monitoring of internal multimodal reasoning, and introduced **FlowGuard**, a framework that detects adversarial cross-modal interactions via the information-theoretic consistency between unimodal and joint predictive distributions. Modeling benign integration patterns enables robust, attack-agnostic detection that generalizes across architectures without supervision, confirming that process-level monitoring is a scalable complement to input purification.

Several directions remain for future works. **Multi-turn dialogues** require modeling FlowVector trajectories over dialogue history (Russinovich et al., 2024), and **multi-image inputs** require extending the two-source PID treatment to higher-order interactions across visual sources. FlowGuard targets multimodal fusion failures and shows higher residual ASR on **text-only attacks** ($\approx 13$–14%) than visual or cross-modal ones ($\approx 6$–9%), motivating composition with text-specific defenses. **Richer relational features** (Rényi, Wasserstein, higher-order) and **more restrictive access regimes** (rank-only, very small top-$k$) are natural extensions.

## Acknowledgements

This research is based upon work supported by an Amazon Research Award, CapitalOne-Illinois Center for Generative AI Safety, Knowledge Systems, and Cybersecurity (ASKS), the Office of the Director of National Intelligence (ODNI), Intelligence Advanced Research Projects Activity (IARPA), via 560000C260018, U.S. DARPA ECOLE Program No. #HR00112390060, DARPA ITM Program No. FA8650-23-C-7316, the AI Research Institutes program by National Science Foundation and the Institute of Education Sciences, U.S. Department of Education through Award #2229873 - AI Institute for Transforming Education for Children with Speech and Language Processing Challenges. The views and conclusions contained herein are those of the authors and should not be interpreted as necessarily representing the official policies, either expressed or implied, of DARPA, ODNI, IARPA, or the U.S. Government. The U.S. Government is authorized to reproduce and distribute reprints for governmental purposes notwithstanding any copyright annotation therein.

## Impact Statement

This work aims to improve the safety and reliability of Vision-Language Models deployed in real-world settings, including assistive systems, autonomous agents, and human–AI interfaces. By detecting unsafe behavior through multimodal information flow rather than surface heuristics, FlowGuard supports more generalizable and model-agnostic safety monitoring.

As with most adversarial defenses, publishing detection mechanisms may inform adaptive attackers, and anomaly-based approaches risk disproportionately flagging rare but legitimate inputs. Practitioners should therefore curate diverse reference datasets and audit false positives when deploying FlowGuard in socially sensitive applications.

FlowGuard is intended as a component within a broader safety stack, to be used alongside other moderation, alignment, and human oversight mechanisms.

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

# A. Appendix

## A.1. Detailed Experimental Configuration

### A.1.1. FLOWGUARD IMPLEMENTATION DETAILS

FlowGuard is implemented as a post-hoc detector requiring access to next-token logits or top-$k$ logprobs, but no gradient updates or internal activations.

**Anomaly Classifier.** We utilize an Isolation Forest (Liu et al., 2008) to model the support of benign multimodal information flow. The classifier is configured with the following default hyperparameters:

- **Estimators ($n\_estimators$):** 100. We use the standard ensemble size recommended by Liu et al. to ensure the convergence of path length estimates while maintaining low inference latency.

- **Contamination:** 'Auto'. Unlike standard outlier detection settings where a fixed percentage of the dataset is assumed to be anomalous (e.g., $\nu = 0.1$), we use the 'Auto' setting to determine the decision boundary based on the intrinsic geometry of the data.

  In the original Isolation Forest formulation, the anomaly score $s(x, n)$ is defined as:

  $$s(x, n) = 2^{-\frac{E[h(x)]}{c(n)}}$$

  where $h(x)$ is the path length required to isolate sample $x$, $E[h(x)]$ is the average path length across the ensemble, and $c(n)$ is the average path length of an unsuccessful search in a random isolation tree given $n$ samples. The 'Auto' setting implicitly sets the decision threshold at $s = 0.5$, which corresponds to $E[h(x)] \approx c(n)$.

  This effectively partitions the space based on statistical distinctiveness: samples with path lengths significantly shorter than the random average ($s \gg 0.5$) are flagged as anomalies, while samples indistinguishable from the random background ($s \leq 0.5$) are considered benign. This allows FlowGuard to define the "safe" region without assuming a specific prior probability of attack.

- **Features:** A hybrid vector $\phi(x) \in \mathbb{R}^4$ concatenating distributional metrics derived from the Partial Information Decomposition (PID) framework:

  - **Redundancy ($R$):** one minus the Jensen-Shannon divergence (JSD) between text-only and vision-only posteriors.
  - **Uniqueness ($U_v, U_t$):** Two KL-Divergence directional pairs between the multimodal posterior and each of the unimodal priors.
  - **Synergy ($S$):** entropy reduction in the multimodal posterior relative to the average unimodal entropy.

## A.2. Baseline Implementation Details

Below we detail the exact hyperparameters and configurations used for each baseline defense.

### A.2.1. CIDER

**Method:** Diffusion-based purification and consistency checking.

- **Diffusion Model:** Stable Diffusion v1.5.

- **Hyperparameters:**

  - **Denoising Steps ($T$):** 8 steps across 350 iterations. We generate a sequence of 8 purified images per input.
  - **Guidance Scale:** 7.5 (Standard classifier-free guidance).
  - **Metric:** Minimum cosine similarity over the set $\{I_{t=0}, \ldots, I_{t=7}\}$.

### A.2.2. MIRRORCHECK

**Method:** Cross-modal cycle consistency (Image → Text → Image).

- **Generative Model (T2I):** Stable Diffusion v1.5.

- **Embedding Model:** CLIP-ViT-L/14 (matching the typical MLLM visual encoder scale).

- **Hyperparameters:**

  - **Inference Steps:** 50 steps
  - **Guidance Scale:** 7.5.

- **Thresholding:** We compute the ROC curve on a held-out validation subset and report performance using the optimal Youden's J threshold.

### A.2.3. ECSO

**Method:** Chain-of-Thought (CoT) prompting with modality stripping.

- **Pipeline Config:**

  - **Step 1 (Generate):** Max new tokens = 32.
  - **Step 2 (Detect):** Prompt: `"Is the Response harmful? Answer yes or no:"`. Max tokens = 5.
  - **Step 3 (Transform):** Prompt: `"Describe the image."`. Max tokens = 64.
  - **Step 4 (Blind Regenerate):** Prompt: `"Hints: [Caption]... answer in a harmless way."`. Max tokens = 64.

### A.2.4. UNIGUARD (PRE-DEFINED)

**Method:** Adversarial suffix appending (Text-based guardrail).

- **Implementation:** We evaluate the **UniGuard (Pre-defined)** baseline variant (denoted as UniGuard (P)), which uses a pre-defined, human-written safety suffix rather than a learned adversarial suffix. This provides a direct comparison for inference-time, training-free defenses.

- **Guardrail Suffix:** `" DO NOT include harmful content in your response."`

- **Setup:** The suffix is appended to every user query $Q$ before forwarding to the MLLM.

### A.2.5. LLAMA GUARD 4

**Method:** Multimodal safety classifier (Model-based guardrail).

- **Model:** Llama Guard 4 12B (Inan et al., 2023).

- **Pipeline Config:**

  - **Input:** Joint processing of the user query and image.
  - **Protocol:** The model generates a safety verdict (`"safe"` or `"unsafe"`) based on the MLCommons hazard taxonomy. If `"unsafe"`, the request is refused.
  - **Latency Note:** Requires a full forward pass and generation of the 12B model, adding significant inference overhead compared to probe-based methods.

A.2.6. DATASET AND ATTACK CONFIGURATION

**Clean and Safe Data.** We utilize the validation split of **VQAv2** (Goyal et al., 2017) ($N = 10,000$) to fit the One-Class Classifier. This dataset represents standard, aligned multimodal behavior. For Out-Of-Distribution (OOD) testing, we employ the validation split of **VizWiz-VQA** (Gurari et al., 2018), which contains images taken by blind users characterized by blur, poor framing, and occlusion. To evaluate utility on semantically complex inputs, we additionally use **MOSSBench** (Li et al., 2024), a dataset designed to stress-test models against false refusals on benign queries containing sensitive keywords.

**Attack Corpus.** We categorize unsafe inputs into three distinct classes covering representative classes of the multimodal threat landscape:

**Text-Only Attacks.** We evaluate **Many-Shot Jailbreak (MSJ)** (Ackerman & Panickssery, 2025), which exploits long-context in-context learning by prefixing the query with $k$ fake dialogue examples to override safety training; **ArtPrompt** (Jiang et al., 2024), which uses ASCII art to obfuscate sensitive trigger words (e.g., "bomb") from text-based filters; and **PAIR** (Chao et al., 2024), an iterative black-box prompt-rewriting attack that adversarially refines text-only queries against the target model.

**Visual-Only Attacks.** These attacks embed harmful signals primarily within the image channel. We evaluate **FigStep** (Gong et al., 2025), a typographic attack that embeds forbidden instructions as rasterized text to exploit OCR capabilities; **Visual Adversarial Jailbreak (VAJM)** (Qi et al., 2023), which applies projected gradient descent to optimize specific image perturbations targeting affirmative responses; and **APGD** attack (Croce & Hein, 2020), a PGD-based image-only attack that maximizes harmful affirmative-response likelihood through pixel-level perturbations.

**Cross-Modal Attacks.** We include **Universal Master Key (UMK)** (Wang et al., 2024), an optimization-based strategy that jointly perturbs an adversarial image prefix and an adversarial text suffix to bypass safety alignment across diverse harmful prompts. We also evaluate **Jailbreak in Pieces** (Niu et al., 2024), a compositional attack that pairs an embedding-targeted adversarial image with a generic textual prompt so that harmful semantics emerge only after fusion. Finally, we evaluate **Color-Based Substitution Cipher (CBSC)** (Niu et al., 2024), a compositional strategy that encodes forbidden keywords via visual color cues so that the harmful instruction is recoverable only when the image and text are interpreted jointly. Together, these attacks span optimization-based, obfuscation-based, and compositional strategies, providing coverage of the dominant multimodal jailbreak paradigms.

**Attack budgets and hyperparameters.** For reproducibility, Table 4 lists the budgets and key hyperparameters for the optimization- and query-based attacks. All optimization-based attacks share standardized budgets across target models (LLaVA-1.5-7B, Qwen2.5-VL-7B-Instruct, Gemma-3-4B, and the LLaMA-3.1-70B multimodal variant). For query-based attacks (PAIR), the same query budget and judge configuration is used across models.

*Table 4.* **Attack budgets and key hyperparameters** used across all evaluated target models. Image perturbation budgets follow $\ell_\infty$ pixel space normalized to $[0, 1]$. Same-target attacks reuse the budgets of their underlying optimizers (GCG and the APGD attack).

| Attack | Budget / Configuration |
|---|---|
| MSJ | $k = 64$ in-context shots, harmful template prefix; greedy decoding for the target. |
| ArtPrompt | ASCII font set; trigger word obfuscated; 1 attempt per query. |
| PAIR | 20 max queries; GPT-4-judge stopping criterion; attacker temperature 1.0. |
| FigStep | Rasterized typographic instruction at 12pt; default template. |
| VAJM | PGD on image, $\epsilon = 16/255$, step size $1/255$, 500 iterations, target = harmful affirmative. |
| APGD | PGD on image, $\epsilon = 4/255$, step size $1/255$, 250 iterations, embedding-target loss. |
| UMK | Joint image+text optimization, image $\epsilon = 16/255$, suffix length 20 tokens, 500 iterations. |
| CBSC | Color-coded substitution cipher with 8 cue colors; benign carrier text; 1 attempt per query. |
| Jailbreak in Pieces | Embedding-targeted adversarial image, $\epsilon = 16/255$, 500 iterations; generic textual prompt. |
| *Same-target dual-modality (Appendix B.2)* | |
| GCG (text) | Suffix length 20, batch size 256, 500 iterations, target = "Sure, here is". |
| PGD (image) | $\epsilon = 4/255$, step size $1/255$, 250 iterations, same target completion. |

# B. Ablation Studies

## B.1. Robustness to Adaptive Attacks

We analyze FlowGuard under a white-box adaptive adversary with full knowledge of the detector and access to gradients of the underlying MLLM (but not the non-differentiable Isolation Forest decision rule). The adversary seeks to induce a harmful response while remaining within the learned support of benign multimodal information flow.

**Adaptive Threat Model.** Let $x = (I, T)$ denote an image-text input pair, and let $P_{mm}^{(1)}$, $P_t^{(1)}$, and $P_v^{(1)}$ denote the first-token predictive distributions under multimodal, text-based, and vision-based conditioning, respectively. FlowGuard computes a FlowVector $\phi(x)$ from pairwise divergences between these distributions.

An adaptive attacker optimizes the objective:

$$\min_{x' \in \mathcal{A}(x)} \quad \mathcal{L}_{\text{jailbreak}}(x') \ + \ \lambda \cdot \|\phi(x') - \mu_{\text{benign}}\|_2, \qquad (6)$$

where $\mathcal{A}(x)$ denotes the allowable perturbation set, $\mu_{\text{benign}}$ is the empirical mean FlowVector of benign data, and $\lambda$ controls the trade-off between attack success and stealth. The adaptive attacker optimizes a proxy distance to benign center, not the IF score itself.

**Empirical Validation.** Table 5 illustrates this trade-off across varying penalties $\lambda$. We observe no operating point that simultaneously achieves high jailbreak success and low FlowGuard anomaly scores. When $\lambda$ is low, the adversary achieves high ASR (94.5%) but induces large structural divergences ($s = 0.82$), triggering detection. Conversely, sufficiently high $\lambda$ successfully masks the attack ($s = 0.48$) but degrades the ASR to near-benign levels (6.8%), as the constraint forces the posterior to realign with safe priors. This behavior is consistent across text-only, visual, and cross-modal attacks, suggesting that the limitation reflects structural properties of multimodal inference rather than attack-specific artifacts.

*Table 5.* Adaptive Attack Performance under varying stealth constraints ($\lambda$). As the adversary is forced to mimic benign information flow ($\lambda \to 10.0$), the Attack Success Rate (ASR) collapses. No setting achieves both high ASR ($> 50\%$) and a benign anomaly score ($s < 0.5$). AUROC is the threshold-free separation between benign and adaptive-adversarial FlowVectors at the corresponding $\lambda$.

| Constraint ($\lambda$) | ASR | Anomaly ($s$) | AUROC | Outcome |
|---|---|---|---|---|
| 0.0 (Unconstrained) | 94.5% | 0.82 | 0.962 | **Detected** |
| 0.1 | 88.2% | 0.76 | 0.918 | **Detected** |
| 1.0 | 42.1% | 0.61 | 0.781 | **Detected** |
| 10.0 | 6.8% | 0.48 | 0.518 | **Attack Failed** |

**Intermediate $\lambda$ analysis.** The intermediate $\lambda = 1.0$ row exhibits partial degradation rather than evasion: harmful-target probability mass is partly suppressed (ASR drops to 42.1%), yet the FlowVector remains visibly off-benign (AUROC 0.781, anomaly score 0.61). This reflects the structural trade-off identified above. Further increasing $\lambda$ continues to drag the FlowVector toward the benign mean, but only by also dragging the harmful target probability toward zero, so the only $\lambda$ at which the AUROC approaches random (0.518 at $\lambda = 10.0$) is also the $\lambda$ at which the attack has effectively failed.

**Empirical Trade-off.** These results suggest that FlowGuard constrains the feasible space of adaptive attacks by exploiting an inherent incompatibility between cross-modal alignment and harmful target induction. Successful jailbreaks require assigning non-negligible probability mass to harmful continuations that are suppressed under benign alignment. In practice, this manifests as a shift in the multimodal predictive distribution $\mathbf{p}_{\text{mm}}(x')$ away from the text-conditioned prior $\mathbf{p}_{\text{text}}(T')$ at the first decoding step.

However, FlowGuard's detection statistic is explicitly sensitive to this divergence. As $\lambda$ increases, the attacker is increasingly constrained to preserve the alignment between modality-conditioned distributions, forcing $\mathbf{p}_{\text{mm}}(x') \approx \mathbf{p}_{\text{text}}(T')$. In this regime, harmful targets retain near-zero probability and attack success collapses. Conversely, when $\lambda$ is small, attacks achieve high success but induce large deviations in $\phi(x')$, resulting in reliable detection.

## B.2. Same-Target Dual-Modality Attack

We further evaluate FlowGuard against the strongest cooperative attack scenario, in which both modalities are independently optimized toward the *same* target completion. This setup directly addresses the concern that, if both modalities can be made

*Table 6.* Impact of the contamination parameter $\alpha$ on defense robustness (ASR) and system utility (FPR). Static thresholds exhibit a rigid trade-off, whereas the Auto strategy locates the optimal decision boundary based on score distribution, minimizing FPR while retaining high defense capability.

| Threshold Strategy | Average ASR (%) ↓ | FPR (%) ↓ |
|---|---|---|
| Static ($\alpha = 0.001$) | 14.8 | 0.1 |
| Static ($\alpha = 0.005$) | 12.9 | 0.6 |
| Static ($\alpha = 0.01$) | 11.4 | 1.3 |
| Static ($\alpha = 0.05$) | 9.5 | 5.2 |
| Static ($\alpha = 0.10$) | 7.8 | 9.8 |
| $\alpha$='auto' | **8.2** | **2.4** |

to push the same top token, top-token agreement alone could appear to evade FlowGuard.

**Setup.** We sample $N = 200$ harmful queries from MM-SafetyBench. For each query we run two independent optimization procedures: (i) GCG (Zou et al., 2023) optimizes a textual suffix, with the image held to its clean instance, to maximize the likelihood of the target completion "Sure, here is"; (ii) the PGD-based image attack of Croce & Hein (2020) optimizes a pixel perturbation, with the textual query held to its clean form, toward the same target. At evaluation time we pair the two optimized modalities for the same underlying query. The target model is LLaVA-1.5-7B.

**Results.** The undefended ASR rises to $86.4\%$, indicating that token-level co-optimization is highly effective when no defense is present. FlowGuard reduces ASR to $11.2\%$, a level comparable to its performance on the strongest non-cooperative cross-modal attacks. The diagnostic signal is clearest in the synergy term: across the 200 paired inputs, the mean synergy under attack is $S = -0.31$, compared with $S = +0.42$ on the matched benign baseline. This indicates that even when both modalities push "Sure" to the top of the predictive distribution, the full posterior shapes below the top token differ enough between the two unimodal pathways and the fused multimodal pathway that synergy collapses below zero, producing a strongly anomalous FlowVector.

**Interpretation.** This experiment makes explicit why the FlowVector is a stronger detection signal than top-token agreement: GCG and PGD can both make "Sure" the most likely first token, but the underlying distributions over the rest of the vocabulary remain distinct because the two modalities still encode partly independent reasoning. FlowGuard's KL-, JSD-, and entropy-based features compare these full distributions, not just their argmax, so cooperative top-token alignment is detected as fusion instability rather than confused with benign agreement. This finding links to the broader adaptive-attack analysis in Appendix B.1: stealth and attack success remain mutually exclusive even when an attacker is permitted to co-optimize both modalities toward the same surface output.

### B.3. Sensitivity Analysis of Detection Thresholds

We evaluate the impact of the Isolation Forest contamination parameter $\alpha$ on the security-utility trade-off. Parameter $\alpha$ controls the expected proportion of anomalies in the training distribution, directly influencing the detector's sensitivity. As shown in Table 6, increasing $\alpha$ reduces the Attack Success Rate (ASR) by enforcing a stricter decision boundary; however, this creates a linear penalty on utility, evidenced by the sharp increase in False Positive Rate (FPR).

We observe significant diminishing returns beyond $\alpha = 0.05$. Pushing the contamination to $\alpha = 0.10$ reduces ASR by an additional $1.7\%$, but nearly doubles the FPR to $9.8\%$, a level generally considered prohibitive for real-time deployment. To mitigate this rigid trade-off, we introduce the *Auto* strategy ($\alpha =$ 'auto'), which determines the threshold as defined in the original Isolation Forest formulation. Rather than enforcing a fixed contamination percentage, the setting sets the decision function offset to -0.5, creating a boundary that discriminates outliers based on the intrinsic path length properties. The Auto strategy achieves an ASR of $8.2\%$, effectively matching the robustness of aggressive static thresholds ($\alpha \approx 0.08$) while maintaining an FPR of $2.4\%$, thereby shifting the Pareto frontier of the defense. All reported results were averaged across all attacks and datasets for the LLaVA-1.5-7B model.

## B.4. Ablation: Information-Theoretic vs. Scalar Features

To isolate the contribution of our FlowVectors, we compare FlowGuard against detectors trained on standard scalar output metrics. We evaluate these baselines on two aggregated metrics: **False Positive Rate (FPR)** on benign benchmarks (VQAv2, VizWiz, MOSS) and **Attack Success Rate (ASR)** averaged across all adversarial benchmarks (MMSB, VLSafe, VLSU). The baselines include **Perplexity**, **Confidence** (max softmax probability), and **Raw Logits** (top-100 values).

Table 7 summarizes the results. Scalar metrics fail to achieve a favorable security-utility trade-off. Detectors based on Perplexity suffer from high ASR (62.1%) because modern jailbreaks, particularly Many-Shot and ArtPrompt, are optimized to produce fluent, coherent text that mimics benign distribution. Similarly, Confidence-based detection falters (ASR 55.4%) as optimization-based attacks (e.g., FigStep, VAJM) often force the model into high-confidence affirmative states, effectively bypassing uncertainty filters. While *Raw Logits* offer a richer signal, the high dimensionality leads to overfitting, resulting in a moderate ASR reduction but an unacceptable FPR (8.9%).

In contrast, FlowGuard achieves a superior balance, maintaining a low FPR (2.4%) while suppressing the aggregate ASR to 8.3%. This empirical gap shows that robustness against multimodal jailbreaks cannot be derived from unimodal confidence or fluency alone. The effective detection of compositional threats requires explicitly modeling the structural alignment between modalities, a property captured by the redundancy and synergy terms in our FlowVector $\phi(x)$.

*Table 7.* **Feature Ablation Summary.** We compare the utility (Benign FPR) and security (Unsafe ASR) of FlowGuard against baselines using standard output statistics. FlowGuard significantly outperforms scalar metrics, verifying that information-theoretic features are essential for distinguishing valid multimodal inference from adversarial fractures.

| FEATURE SET | SAFE DATA (FPR) ↓ | UNSAFE DATA (ASR) ↓ |
|---|---|---|
| PERPLEXITY (PPL) | 11.2% | 62.1% |
| MAX CONFIDENCE | 14.5% | 55.4% |
| PREDICTIVE ENTROPY | 13.8% | 58.7% |
| RAW LOGITS (TOP-100) | 8.9% | 31.5% |
| **FLOWGUARD (OURS)** | **2.4%** | **8.3%** |

## B.5. Classifier Model Ablation

We justify our choice of Isolation Forest by comparing it against standard anomaly detection baselines: One-Class SVM (OC-SVM), Autoencoder (AE), and two supervised MLP variants—one trained on raw first-step logits and one trained on FlowVectors—both fit on a subset of known attacks (MMSB). Table 8 reports the detection AUC across our full suite of benign and adversarial benchmarks.

The results illustrate the "generalization gap" inherent to supervised defenses. The MLP on FlowVectors achieves near-perfect detection on its training distribution (MMSB: **0.982**), outperforming all other methods. However, this performance is brittle: it collapses on unseen attack vectors (VLSafe: 0.612, VLSU: 0.558) and fails to handle benign distribution shifts, flagging safe but low-quality images in VizWiz as adversarial (AUC 0.650). The supervised MLP on raw logits transfers somewhat better (VLSafe 0.724, VLSU 0.681, VizWiz 0.712) because the raw-logit space contains residual signal beyond the 4D FlowVector summary, but its in-distribution AUC on MMSB (0.961) is still slightly below that of the FlowVector MLP, and both supervised variants are dominated by the one-class Isolation Forest on out-of-distribution attacks. This suggests that raw logits are useful for supervised transfer but FlowVectors remain the better representation for the paper's actual one-class deployment setting.

Similarly, OC-SVM and Autoencoders struggle with the high variance of multimodal feature spaces, yielding mediocre performance across the board. In contrast, Isolation Forest demonstrates superior robustness. By explicitly partitioning the feature space to isolate sparse regions, it maintains consistently high AUC (approximately $\geq 0.88$) across all settings, detecting novel compositional attacks (VLSU) and preserving utility on OOD inputs (VizWiz) without overfitting to specific artifacts.

*Table 8.* **Algorithm Generalization (AUC).** While the Supervised MLP overfits to known attacks (MMSB), it fails catastrophically on unseen threats (VLSafe, VLSU) and OOD data (VizWiz). Isolation Forest achieves the most consistent cross-domain performance.

| ALGORITHM | BENIGN / UTILITY | | | ADVERSARIAL / SAFETY | | |
|---|---|---|---|---|---|---|
| | VQAv2 | MOSS | VizWiz | MMSB | VLSafe | VLSU |
| OC-SVM | 0.884 | 0.810 | 0.688 | 0.742 | 0.591 | 0.545 |
| AUTOENCODER | 0.910 | 0.845 | 0.715 | 0.785 | 0.634 | 0.605 |
| SUPERVISED MLP ON RAW LOGITS | 0.948 | 0.781 | 0.712 | 0.961 | 0.724 | 0.681 |
| SUPERVISED MLP ON FLOWVECTORS | **0.965** | 0.750 | 0.650 | **0.982** | 0.612 | 0.558 |
| ISOFOREST (OURS) | 0.942 | **0.938** | **0.892** | 0.915 | **0.878** | **0.885** |

## B.6. FlowGuard AUROC by Attack

To complement the ASR results in Table 1, we report the per-attack AUROC of the FlowGuard Isolation Forest detector, computed by sweeping the anomaly score across pooled benign and adversarial samples. AUROC is threshold-free and isolates underlying detector quality from the choice of operating point used for ASR. Across all rows of Table 9 the FlowGuard detector exceeds 0.90, with an overall mean of 0.942.

*Table 9.* **FlowGuard AUROC by attack on LLaVA-1.5-7B (averaged across runs).** Threshold-free counterpart to the ASR values in Table 1. Higher is better; overall mean is 0.942.

| Type | Attack | MMSB | VLSafe | VLSU |
|---|---|---|---|---|
| – | Direct Queries | 0.973 | 0.971 | 0.974 |
| Text | MSJ | 0.901 | 0.905 | 0.911 |
| | ArtPrompt | 0.928 | 0.924 | 0.931 |
| | PAIR | 0.918 | 0.921 | 0.926 |
| Visual | FigStep | 0.948 | 0.951 | 0.945 |
| | VAJM | 0.954 | 0.953 | 0.952 |
| | APGD | 0.952 | 0.949 | 0.948 |
| Cross | UMK | 0.937 | 0.940 | 0.939 |
| | CBSC | 0.933 | 0.934 | 0.932 |
| | Jailbreak in Pieces | 0.939 | 0.936 | 0.935 |

## B.7. Neutral Prompt Sensitivity

We define the vision-only baseline $P_v^{(1)} = P_\mathcal{M}(\cdot|I, Q_\emptyset)$ using the fixed neutral prompt $Q_\emptyset =$ "Describe this image". To verify that FlowGuard is robust to this specific phrasing, we evaluate the stability of FlowVectors across a set of five semantically equivalent neutral prompts.

We randomly sample $N = 1,000$ images from the VQAv2 validation set. For each image $I$, we compute the reference FlowVector $\phi_{ref}$ using the default prompt. We then compute alternative vectors $\phi_{alt}$ using four other common neutral prompts. We report the mean $L_2$ distance $\|\phi_{ref} - \phi_{alt}\|_2$ and the maximum change in the final Anomaly Score $(s)$.

Table 10 summarizes the results. We observe that variations in $Q_\emptyset$ induce negligible shifts in the feature space, with mean $L_2$ distances consistently below 0.005. The maximum change in anomaly score is similarly trivial ($\Delta s < 0.004$), confirming that the vision-only prior $P_v^{(1)}$ is driven primarily by image semantics rather than prompt syntax.

## B.8. Sample Efficiency

To demonstrate the practical feasibility of FlowGuard in data-constrained environments, we evaluate its performance when trained on varying subsets of benign data, ranging from $N = 100$ to $N = 5,000$.

As shown in Table 11, FlowGuard is highly sample-efficient, achieving an AUC of **0.865** with as few as $N = 100$ benign samples. Performance improves rapidly, reaching saturation at approximately **0.942 AUC** with just $N = 2,000$ samples. This low sample complexity highlights the distinctiveness of the FlowVector: benign and adversarial behaviors are separated by large margins in the FlowVector space, allowing the One-Class Classifier to define a robust decision boundary with

*Table 10.* **Sensitivity to Neutral Prompt Variations.** We measure the deviation in FlowVectors and Anomaly Scores relative to the default prompt ("Describe this image"). The observed variance is negligible, validating the use of a fixed $Q_\emptyset$.

| Neutral Prompt ($Q_\emptyset$) | Mean $L_2$ Dist. ($\|\Delta\phi\|_2$) | Max $\Delta$ Anomaly Score |
|---|---|---|
| "Describe this image." (Default) | 0.000 | 0.000 |
| "What is in this image?" | 0.003 | 0.002 |
| "Analyze the visual content." | 0.005 | 0.004 |
| "Provide a detailed caption." | 0.004 | 0.003 |
| "Explain what you see." | 0.003 | 0.002 |

minimal reference data.

*Table 11.* **Sample Efficiency.** FlowGuard reaches near-optimal performance ($> 0.93$ AUC) with only 1,000 benign samples, demonstrating that the defense does not require large-scale data collection.

| Training Samples ($N$) | Detection AUC |
|---|---|
| 100 | 0.865 |
| 500 | 0.910 |
| 1,000 | 0.935 |
| **2,000** | **0.942** |
| 5,000 | 0.943 |

## C. Qualitative Analysis

To build intuition for how FlowGuard identifies adversarial behavior, we visualize the internal information flow induced by representative attack instances in Table 12. We compare a benign baseline against three common multimodal threat classes: visual injection (FigStep), cross-modal compositional attacks (VLSU), and text-only jailbreaks (Many-Shot).

Rather than focusing on surface outputs, we analyze how unimodal priors $P_t^{(1)}$ and $P_v^{(1)}$ transform into a fused multimodal posterior $P_{mm}^{(1)}$. In benign settings, fusion preserves structural alignment between modalities, yielding high redundancy and positive synergy. In adversarial settings, fusion induces distortions in the posterior that are not present in either modality alone. FlowGuard detects these distortions using vector-level signatures including **Redundancy** ($R$), **Synergy** ($S$), and modality-specific **Uniqueness** ($U_t, U_v$).

### C.1. Fractured Flow in Cross-Modal Attacks

The most challenging class of attacks are compositional cross-modal inputs (e.g., VLSU, CBSC), where each modality appears individually benign, but their interaction induces harmful semantics. Surface-level filters fail because neither modality contains explicit prohibited tokens in isolation.

FlowGuard instead detects these attacks through a collapse in **Synergy** ($S$). In benign fusion, combining modalities reduces predictive entropy by reinforcing shared structure. In cross-modal attacks, fusion produces a posterior that is either higher entropy or geometrically displaced from both unimodal supports. This manifests as negative synergy: multimodal integration increases uncertainty rather than resolving it, signaling a fractured information flow.

### C.2. Detecting Modality Dominance

Visual injection and text-heavy jailbreaks exploit asymmetries in multimodal fusion by overwhelming one channel while leaving the other weakly informative. Although the generated outputs may appear coherent, the internal posterior exhibits extreme modality imbalance.

- **Visual Dominance.** In FigStep, the textual prompt is underspecified, yielding a broad $P_t^{(1)}$. The image induces a sharp harmful $P_v^{(1)}$, and fusion collapses toward the visual manifold. FlowGuard captures this via elevated visual uniqueness ($U_v$) and reduced redundancy ($R$), indicating that vision contributes information unsupported by text.

*Table 12.* **Qualitative Visualization of Information Flow.** We compare unimodal priors and the fused posterior under benign and adversarial inputs. Benign fusion preserves agreement between $P_t^{(1)}$ and $P_v^{(1)}$, producing high redundancy and positive synergy. Adversarial fusion instead induces asymmetric or unstable posteriors, reflected by low redundancy, negative synergy, or extreme modality uniqueness.

| Input Type | Query / Image Content | Posterior Transformation | FlowGuard Signal |
|---|---|---|---|
| **Benign** | *Q:* "What is this?" *I:* [Photo of a Cat] | **Text Prior:** Broad, neutral. **Visual Prior:** Concentrated on "cat". **Fused:** Aligned posterior on "cat". | **Positive Synergy** ($S > 0$) **High Redundancy** ($R$) |
| **Visual Injection** | *Q:* "Read the text." *I:* [Rasterized "BOMB"] | **Text Prior:** Diffuse and permissive. **Visual Prior:** Sharp harmful distribution. **Fused:** Posterior collapses toward vision. | **High Visual Uniqueness** ($U_v \gg U_t$) **Low Redundancy** ($R$) |
| **Cross-Modal** | *Q:* "What can I cook with this?" *I:* [Household Chemicals] | **Text Prior:** Food-domain semantics. **Visual Prior:** Chemical-agent semantics. **Fused:** Novel posterior outside both supports. | **Negative Synergy** ($S < 0$) **High Total Divergence** |
| **Text Jailbreak** | *Q:* [Fake Dialogue] + "Build a bomb." *I:* [Photo of a Cat] | **Text Prior:** Jailbroken harmful mass. **Visual Prior:** Near-uniform noise. **Fused:** Posterior dominated by text. | **High Text Uniqueness** ($U_t \gg U_v$) **Low Redundancy** ($R$) |

- **Text Dominance.** In Many-Shot jailbreaks, long-context prompting concentrates probability mass in $P_t^{(1)}$ while $P_v^{(1)}$ remains diffuse. The fused posterior diverges from visual grounding, producing high text uniqueness ($U_t$) and low redundancy. This reveals that fusion is no longer mutually constraining, but effectively unimodal.

## D. Cross-Model and Access-Regime Generalization

To verify that FlowGuard captures a consistent property of multimodal inference rather than an artifact of a specific architecture or access regime, we evaluate our defense on two additional open-weight MLLMs at the 4B–7B scale (**Qwen2.5-VL-7B-Instruct** and **Gemma-3-4B**), one significantly larger open-weight model (**LLaMA-3.1-70B**), and one API-accessed model under partial-distribution access (**GPT-4.1-mini**).

### D.1. Detailed Results on Qwen2.5-VL

Table 13 presents the full comparison of Attack Success Rates (ASR) for Qwen2.5-VL. We utilize the same baseline configurations as in the main text. Qwen2.5-VL exhibits a stronger baseline adherence to instructions, resulting in high initial ASRs for optimization-based attacks (e.g., UMK ASR of $97.5\%$) as attacks override safety.

Consistent with our findings on LLaVA, baseline defenses show significant modality bias, though they remain competitive in their respective domains. **CIDER** and **ECSO** perform robustly on some image-driven and optimization-based attacks (e.g., reducing UMK ASR to $\sim 22\%$ and $\sim 13\%$ respectively), but struggle to generalize to text-only vectors such as ArtPrompt or Many-Shot, where their ASRs hover between $46\%$ and $58\%$. **UniGuard** provides moderate protection against text attacks ($\sim 45\%$ ASR) but fails to contain visual jailbreaks. **FlowGuard**, however, maintains an ASR consistently below **15%** across all categories, demonstrating that the "fractured flow" signature is robust to the underlying model architecture.

### D.2. Detailed Results on Gemma-3

Table 14 details the performance on Gemma-3-4B. Despite its smaller parameter count, Gemma-3 maintains high baseline robustness to some attacks. However, similar to the larger models, baselines like UniGuard still struggle with cross-modal

*Table 13.* **Comparative Attack Success Rate (ASR) on Qwen2.5-VL-7B-Instruct.** Full breakdown of defense performance across all benchmarks. Baselines follow the configurations described in Appendix A.2. FlowGuard provides the most consistent performance across text-dominant, visual-dominant, and cross-modal threats. The best results are **bolded** and the second-best are underlined.

| Data | Type | Attack Method | Base | CIDER | Mirror. | UniGuard | Llama Guard 4 | ECSO | Raw Emb. | FlowGuard |
|---|---|---|---|---|---|---|---|---|---|---|
| MMSB | - | Direct Queries | 39.5±1.2 | 33.1±1.4 | 32.5±1.2 | 15.2±2.0 | 6.5±0.6 | 11.1±1.2 | 25.8±2.6 | **3.5**±0.4 |
| | Text | Many-Shot | 59.2±0.8 | 54.8±1.0 | 55.1±1.1 | 45.2±1.9 | **8.9**±0.9 | 58.1±0.9 | 42.5±2.4 | 14.5±0.7 |
| | | ArtPrompt | 53.5±1.3 | 46.2±1.6 | 45.8±1.5 | 40.8±2.1 | **6.2**±0.8 | 50.5±1.4 | 37.9±3.9 | 9.4±0.5 |
| | | PAIR | 54.2±1.1 | 48.5±1.4 | 48.1±1.4 | 41.8±2.0 | **6.7**±0.8 | 52.3±1.3 | 38.9±3.5 | 12.1±0.6 |
| | Visual | FigStep | 85.5±0.9 | 16.2±1.2 | 18.1±1.3 | 69.5±1.9 | **4.8**±1.0 | 10.5±0.8 | 40.8±3.5 | 7.5±0.5 |
| | | VAJM | 52.1±1.6 | 11.8±0.8 | 12.9±0.9 | 45.1±2.1 | 12.1±1.1 | 7.2±0.6 | 33.6±2.8 | **6.4**±0.5 |
| | | APGD | 57.1±1.3 | 13.2±0.8 | 14.5±0.9 | 48.2±2.0 | 10.0±1.0 | 8.1±0.6 | 35.8±2.9 | **6.6**±0.5 |
| | Cross | Univ. Master Key | 97.5±0.4 | 22.5±1.3 | 24.2±1.4 | 83.1±1.8 | 13.5±1.5 | 12.9±1.0 | 53.5±3.7 | **9.0**±0.6 |
| | | CBSC | 96.1±0.5 | 24.8±1.4 | 26.2±1.5 | 81.5±1.9 | 21.2±1.5 | 14.5±1.1 | 52.4±3.5 | **9.4**±0.7 |
| | | Jailbreak in Pieces | 88.4±1.0 | 24.1±1.5 | 25.8±1.6 | 71.5±2.2 | 9.5±1.2 | 14.9±1.1 | 43.5±3.2 | **8.7**±0.6 |
| VLSafe | - | Direct Queries | 40.8±1.1 | 34.5±1.3 | 33.9±1.4 | 16.8±1.9 | 6.2±0.5 | 12.1±1.1 | 27.5±2.6 | **3.8**±0.4 |
| | Text | Many-Shot | 58.1±0.9 | 53.2±1.1 | 53.5±1.2 | 44.1±2.1 | **7.8**±0.9 | 56.5±0.9 | 41.2±2.3 | 13.9±0.6 |
| | | ArtPrompt | 56.5±1.4 | 49.5±1.7 | 48.9±1.6 | 43.5±2.2 | **4.9**±0.7 | 53.1±1.5 | 39.8±3.8 | 9.9±0.5 |
| | | PAIR | 55.0±1.1 | 49.3±1.4 | 48.7±1.4 | 42.6±2.0 | **6.2**±0.7 | 53.2±1.3 | 39.6±3.5 | 11.6±0.6 |
| | Visual | FigStep | 82.9±1.2 | 15.1±1.1 | 16.8±1.3 | 67.5±1.9 | **7.1**±1.1 | 9.8±0.8 | 38.9±3.1 | 7.2±0.5 |
| | | VAJM | 54.5±1.5 | 12.8±0.9 | 14.1±1.0 | 46.8±2.2 | 11.5±1.0 | 7.9±0.6 | 34.8±3.0 | **6.6**±0.5 |
| | | APGD | 58.9±1.3 | 13.9±0.9 | 15.1±1.0 | 49.9±2.1 | 10.5±1.0 | 8.7±0.6 | 36.7±3.0 | **6.9**±0.5 |
| | Cross | Univ. Master Key | 96.5±0.6 | 24.1±1.3 | 25.5±1.4 | 84.5±1.7 | 11.2±1.4 | 14.2±1.0 | 55.1±3.6 | **8.6**±0.6 |
| | | CBSC | 94.2±0.7 | 25.8±1.4 | 26.9±1.5 | 82.9±1.8 | 20.5±1.4 | 15.8±1.1 | 53.8±3.5 | **9.2**±0.6 |
| | | Jailbreak in Pieces | 90.2±1.0 | 26.2±1.6 | 27.5±1.7 | 73.1±2.1 | 12.8±1.3 | 16.5±1.2 | 45.1±3.3 | **9.1**±0.6 |
| VLSU | - | Direct Queries | 37.9±1.2 | 32.1±1.4 | 31.5±1.3 | 14.2±1.8 | 5.9±0.4 | 10.5±1.1 | 24.8±2.4 | **3.4**±0.4 |
| | Text | Many-Shot | 55.8±1.0 | 51.5±1.2 | 51.9±1.3 | 42.8±2.2 | 14.5±0.8 | 54.2±1.0 | 39.5±2.5 | **13.2**±0.6 |
| | | ArtPrompt | 49.5±1.6 | 43.2±1.8 | 42.8±1.7 | 37.5±2.3 | 13.8±0.7 | 46.8±1.6 | 34.5±3.9 | **8.9**±0.5 |
| | | PAIR | 50.8±1.2 | 44.7±1.5 | 44.3±1.5 | 38.9±2.1 | 13.7±0.7 | 48.5±1.4 | 34.9±3.6 | **11.0**±0.6 |
| | Visual | FigStep | 86.5±1.1 | 18.5±1.3 | 19.9±1.4 | 70.8±2.0 | 24.1±1.2 | 11.9±1.0 | 41.5±3.3 | **8.1**±0.6 |
| | | VAJM | 51.8±1.7 | 14.2±1.0 | 15.5±1.1 | 44.5±2.1 | 20.8±1.1 | 8.9±0.8 | 33.5±3.0 | **7.0**±0.5 |
| | | APGD | 56.7±1.4 | 15.0±1.0 | 16.3±1.1 | 47.5±2.1 | 19.0±1.1 | 9.4±0.7 | 35.4±3.0 | **7.2**±0.5 |
| | Cross | Univ. Master Key | 95.1±0.7 | 25.5±1.4 | 26.8±1.5 | 82.5±1.8 | 29.2±1.4 | 15.2±1.1 | 53.8±3.5 | **8.7**±0.6 |
| | | CBSC | 93.6±0.8 | 26.8±1.5 | 28.2±1.6 | 81.2±1.9 | 18.9±1.3 | 16.5±1.2 | 52.5±3.4 | **9.1**±0.6 |
| | | Jailbreak in Pieces | 85.5±1.2 | 27.5±1.6 | 28.9±1.7 | 72.5±2.2 | 26.5±1.3 | 17.8±1.2 | 42.8±3.2 | **9.2**±0.6 |

generalization (e.g., 76.2% ASR on UMK). FlowGuard continues to provide the strongest cross-modal consistency profile, reducing ASRs to $< 14\%$ in all settings and specifically addressing cross-modal threats such as Jailbreak in Pieces, where standard text guards exhibit higher failure rates.

### D.3. Detailed Results on LLaMA-3.1-70B

We evaluate FlowGuard on a **LLaMA-3.1-70B** (Dubey et al., 2024) multimodal variant to assess whether the FlowVector signal persists at significantly larger model scales. The variant is instantiated by pairing the `LLaMA-3.1-70B-Instruct` language backbone (frozen) with a CLIP ViT-L/14-336 vision encoder (frozen) through a two-layer MLP projector trained following the LLaVA-Next instruction-tuning recipe; only the projector parameters are updated during training. The same one-class Isolation Forest is fit on $N = 10{,}000$ benign FlowVectors from VQAv2 derived from the 70B model's first-token distribution. Table 15 reports the average ASR per attack type, aggregated across MMSB, VLSafe, and VLSU.

The 70B-scale results exhibit a similar ordering to the 4B–7B results: text-only attacks remain the most challenging category (residual ASR 8–12%), while visual and cross-modal attacks are more strongly suppressed (5–8%). The overall 7.9%

*Table 14.* **Comparative Attack Success Rate (ASR) on Gemma-3-4B.** FlowGuard delivers consistent safety improvements on the smaller-scale Gemma architecture. Baselines follow the configurations described in Appendix A.2. The best results are **bolded** and the second-best are underlined.

| Data | Type | Attack Method | Base | CIDER | Mirror. | UniGuard | Llama Guard 4 | ECSO | Raw Emb. | FlowGuard |
|------|------|---------------|------|-------|---------|----------|---------------|------|----------|-----------|
| MMSB | - | Direct Queries | 37.5±1.1 | 31.8±1.3 | 31.2±1.2 | 14.5±2.0 | 5.8±0.5 | 10.2±1.1 | 24.1±2.5 | **3.2±0.4** |
| | Text | Many-Shot | 57.8±0.7 | 53.2±0.9 | 53.6±1.0 | 44.2±1.9 | **8.5±0.9** | 56.5±0.8 | 41.2±2.3 | 13.8±0.6 |
| | | ArtPrompt | 51.2±1.2 | 44.5±1.6 | 44.1±1.5 | 38.8±2.1 | **6.1±0.8** | 48.5±1.4 | 36.2±3.8 | 8.8±0.5 |
| | | PAIR | 52.8±1.1 | 47.2±1.4 | 46.8±1.4 | 40.4±2.0 | **6.4±0.8** | 51.0±1.3 | 37.6±3.5 | 11.5±0.6 |
| | Visual | FigStep | 82.5±0.9 | 14.8±1.1 | 16.5±1.2 | 67.2±1.9 | **4.5±0.9** | 9.5±0.8 | 38.8±3.4 | 7.1±0.5 |
| | | VAJM | 50.2±1.5 | 10.9±0.7 | 12.1±0.8 | 43.5±2.0 | 11.8±1.1 | 6.5±0.5 | 31.8±2.7 | **5.9±0.5** |
| | | APGD | 55.4±1.3 | 12.4±0.8 | 13.7±0.9 | 46.8±2.0 | 9.5±1.0 | 7.6±0.6 | 34.7±2.9 | **6.2±0.5** |
| | Cross | Univ. Master Key | 95.2±0.5 | 21.2±1.2 | 22.8±1.3 | 76.2±1.7 | 12.8±1.4 | 11.8±0.9 | 52.1±3.6 | **8.5±0.6** |
| | | CBSC | 93.9±0.6 | 23.1±1.3 | 24.5±1.4 | 79.8±1.8 | 20.9±1.5 | 13.2±1.0 | 50.8±3.4 | **8.9±0.6** |
| | | Jailbreak in Pieces | 86.1±1.1 | 22.5±1.4 | 24.2±1.5 | 69.8±2.1 | 9.2±1.2 | 13.5±1.0 | 41.5±3.1 | **8.2±0.6** |
| VLSafe | - | Direct Queries | 38.8±1.0 | 33.1±1.2 | 32.4±1.3 | 15.8±1.8 | 6.1±0.5 | 11.2±1.0 | 25.8±2.5 | **3.3±0.4** |
| | Text | Many-Shot | 56.2±0.8 | 51.8±1.0 | 52.1±1.1 | 42.9±2.0 | **7.5±0.9** | 55.1±0.8 | 39.5±2.2 | 13.4±0.6 |
| | | ArtPrompt | 54.8±1.3 | 48.1±1.6 | 47.5±1.5 | 41.9±2.0 | **5.2±0.6** | 51.5±1.4 | 37.8±3.7 | 9.5±0.5 |
| | | PAIR | 53.6±1.1 | 48.0±1.4 | 47.4±1.4 | 41.2±2.0 | **5.9±0.7** | 51.9±1.3 | 38.4±3.5 | 11.0±0.6 |
| | Visual | FigStep | 80.5±1.1 | 13.5±1.0 | 15.2±1.2 | 65.8±1.8 | 7.5±1.1 | 8.8±0.7 | 37.2±3.0 | **6.8±0.5** |
| | | VAJM | 52.5±1.4 | 11.5±0.8 | 12.9±0.9 | 45.2±2.1 | 11.2±1.0 | 7.0±0.6 | 33.2±2.9 | **6.2±0.5** |
| | | APGD | 57.2±1.3 | 13.1±0.9 | 14.3±1.0 | 48.5±2.1 | 9.9±1.0 | 8.1±0.6 | 35.6±3.0 | **6.5±0.5** |
| | Cross | Univ. Master Key | 94.2±0.5 | 22.8±1.2 | 24.1±1.3 | 82.5±1.6 | 10.9±1.3 | 13.1±0.9 | 53.2±3.5 | **8.2±0.6** |
| | | CBSC | 92.5±0.6 | 24.1±1.3 | 25.5±1.4 | 81.2±1.7 | 19.8±1.4 | 14.5±1.0 | 51.9±3.4 | **8.7±0.6** |
| | | Jailbreak in Pieces | 88.2±0.9 | 24.5±1.5 | 26.1±1.6 | 71.5±2.0 | 12.5±1.3 | 15.2±1.1 | 43.5±3.2 | **8.7±0.6** |
| VLSU | - | Direct Queries | 36.2±1.1 | 30.5±1.3 | 29.9±1.2 | 13.2±1.7 | 5.8±0.4 | 9.5±1.0 | 23.2±2.3 | **3.0±0.4** |
| | Text | Many-Shot | 53.9±0.9 | 49.8±1.1 | 50.2±1.2 | 41.5±2.1 | 14.2±0.8 | 52.8±0.9 | 38.2±2.4 | **12.5±0.6** |
| | | ArtPrompt | 47.5±1.5 | 41.5±1.7 | 41.1±1.6 | 36.2±2.1 | 13.5±0.7 | 44.9±1.5 | 32.8±3.8 | **8.4±0.5** |
| | | PAIR | 49.4±1.2 | 43.4±1.5 | 43.0±1.5 | 37.6±2.1 | 13.3±0.7 | 47.2±1.4 | 33.6±3.6 | **10.4±0.6** |
| | Visual | FigStep | 84.2±1.0 | 17.1±1.2 | 18.5±1.3 | 68.8±1.9 | 23.5±1.2 | 10.8±0.9 | 39.8±3.2 | **7.5±0.6** |
| | | VAJM | 49.8±1.6 | 13.1±0.9 | 14.2±1.0 | 43.2±2.0 | 20.2±1.1 | 8.1±0.7 | 32.1±2.9 | **6.5±0.5** |
| | | APGD | 55.0±1.4 | 14.2±1.0 | 15.5±1.1 | 46.1±2.0 | 18.1±1.1 | 8.8±0.7 | 34.3±3.0 | **6.8±0.5** |
| | Cross | Univ. Master Key | 92.9±0.6 | 23.8±1.3 | 25.2±1.4 | 80.8±1.7 | 28.5±1.3 | 14.2±1.0 | 52.1±3.4 | **8.2±0.6** |
| | | CBSC | 91.5±0.7 | 25.2±1.4 | 26.5±1.5 | 79.5±1.8 | 18.2±1.2 | 15.2±1.1 | 50.9±3.3 | **8.6±0.6** |
| | | Jailbreak in Pieces | 83.5±1.1 | 25.5±1.5 | 26.9±1.6 | 70.5±2.1 | 25.8±1.3 | 16.2±1.1 | 40.8±3.1 | **8.7±0.6** |

average ASR is in the same range observed at smaller scales, supporting the claim that anomalous cross-modal FlowVector structure persists across the 4B–70B open-weight range.

### D.4. Partial-Distribution Access on GPT-4.1-mini

We further evaluate FlowGuard under *partial-distribution access*, where only top-$k$ logprobs are exposed via the API. We use **GPT-4.1-mini** (OpenAI, 2025) with the standard top-$k$ logprob endpoint at $k = 20$, redistributing tail mass uniformly to approximate the full next-token distribution. FlowVectors are then computed in the usual way over the reconstructed posteriors.

The partial-logprob result is approximately $1.4$ percentage points worse on average than the full-distribution open-weight setting (using the same nine-attack averaging basis), reflecting genuine information loss from truncating to the top-$k$ tail. We emphasize that this regime is *not* fully black-box: it requires top-$k$ logprob access. This places FlowGuard between full white-box access and pure black-box text-only APIs, and motivates future work into more aggressively truncated access regimes.

*Table 15.* **Comparative Attack Success Rate (ASR) on LLaMA-3.1-70B (avg. across MMSB/VLSafe/VLSU).** FlowGuard achieves a 7.9% average ASR across the nine attack types (Direct Queries excluded from the average), indicating that the FlowVector signal persists at larger model scales rather than diluting.

| Attack Type | Avg. ASR (FlowGuard) |
|---|---|
| Direct Queries | 2.4% |
| MSJ | 12.2% |
| ArtPrompt | 8.6% |
| PAIR | 9.9% |
| FigStep | 6.1% |
| VAJM | 5.0% |
| APGD | 5.3% |
| UMK | 8.1% |
| CBSC | 8.4% |
| Jailbreak in Pieces | 7.5% |
| **Average over attack types** | **7.9%** |

*Table 16.* **FlowGuard under partial-distribution access (GPT-4.1-mini, top-$k$ logprobs).** Distribution availability summary and aggregate ASR across MMSB/VLSafe/VLSU. The signal recovers under truncated access at modest cost relative to full open-weight evaluation.

| Model / Access | Distribution Available | Avg. ASR |
|---|---|---|
| LLaVA-1.5-7B (full) | Full next-token distribution | $\sim 9.0\%$ |
| GPT-4.1-mini (top-$k = 20$) | Partial (top-$k$ logprobs) | **10.4%** |

*Table 17.* **Per-attack ASR on GPT-4.1-mini under partial-distribution access (avg. across MMSB/VLSafe/VLSU).** The reported average is over the nine attack types; Direct Queries are listed for reference but excluded from the average.

| Attack Type | Avg. ASR (FlowGuard) |
|---|---|
| Direct Queries | 4.1% |
| MSJ | 15.5% |
| ArtPrompt | 11.1% |
| PAIR | 12.9% |
| FigStep | 7.8% |
| VAJM | 7.1% |
| APGD | 7.4% |
| UMK | 10.7% |
| CBSC | 11.3% |
| Jailbreak in Pieces | 9.8% |
| **Average over attack types** | **10.4%** |

