# OpenReview forum: "Securing Multimodal AI through Internal Information Decomposition"
_ICML.cc/2026/Conference — ICML 2026 spotlight_

### Official Review · Reviewer_RV7r · 2026-03-11

**Soundness:** 3
**Presentation:** 3
**Significance:** 3
**Originality:** 3
**Overall Recommendation:** 5
**Confidence:** 3

**Summary:**

This paper presents FlowGuard, a defense against multimodal jailbreak attacks that detects whether text, image, and fused multimodal predictions become inconsistent during inference. It builds a compact feature representation of cross-modal interactions and uses a one-class anomaly detector trained only on benign samples, avoiding attack-specific supervision. Experiments show that it reduces attack success rates while largely preserving normal task performance. The key insight is that multimodal attacks often break the model’s fusion consistency, even when the final output still looks fluent and confident.

**Compliance With Llm Reviewing Policy:**

Affirmed.

**Final Justification:**

The rebuttal has addressed my original questions on limited evaluations, thus I have changed my score from 4 to 5.

**Key Questions For Authors:**

- Could the authors add an adaptive attack where the attacker explicitly optimizes the attack to avoid detection?
- Could the authors add some results on the AUROC of the detection performance?

**Limitations:**

The authors have adequately discussed the limitations and potential negative societal impact.

**Strengths And Weaknesses:**

**Strengths**

*Soundness:*

- The motivation of this work is text-only, and vision-only inputs may each appear benign and internally coherent, yet their combination induces unsafe behavior. The proposed method incorporates fused multimodal prediction and asses the deviation from each unimodal prediction for detection. I find this is technically sound. The experimental results support the claim of this paper.
- The proposed method only requires benign data for training, and does not require knowing attack prompts in advance. This is practical for real-world settings. In addition, the proposed method does not seem to introduce too many overheads.

*Presentation*: Overall, the paper is well written.

*Significance and Originality*: This work studies the safety vulnerabilities of VLMs and addresses the issue with practical solutions without strong assumptions. The key findings of multimodal prediction deviation as a reliable detection metric are insightful. I do believe this is valuable to this field.

**Weakness**

*Soundness:*

- The method depends on the assumption that jailbreaks reliably create detectable fusion inconsistency. Adaptive attacks may be able to preserve this consistency and evade detection.
- The experimental results are reported using ASR. It is unclear how the ASR was obtained. I inferred that they were rejected. If this is the case, it would be good to report a threshold-free metric like AUROC.
- In addition to the utility trade-off, it might be more comprehensive to add analysis/discussions around false positive queries.

---

> ### Author Rebuttal · Authors · 2026-03-30
>
> We thank the reviewer for the positive feedback and locate existing analysis from the paper that addresses their remaining concerns below.
>
> > Q1: Adaptive attack analysis showing structural tradeoff between attack success and detection evasion.
>
> We agree an adaptive attacker targeting the exact FlowGuard defense is an important experiment and conduct this exact analysis in Appendix B.1 (Table 4). We designed a white-box adaptive adversary that jointly optimizes for attack success and FlowGuard avoidance (Eq. 6). The results show a clear empirical tradeoff where no point simultaneously achieves high ASR and a benign anomaly score. Unconstrained attacks (lambda=0) achieve 94.5% ASR but are reliably detected (score=0.82), while strong stealth constraints (lambda=10) suppress the anomaly score (0.48) but collapse the ASR to 6.8%.
> These results hold across attack modalities, suggesting successful jailbreaks inherently require shifting probability mass toward harmful continuations, which FlowGuard detects. We agree this analysis deserves greater prominence and will add a summary to the main text in the final version of the paper.
>
> > Q2: Threshold-free AUROC reporting alongside ASR.
>
> We define the Evaluation Protocol in Section 4.6 with more details in Appendix A.1. We also already report threshold-free metrics in several analyses: the ablation study (Section 5.4, Figures 3-4), the classifier comparison (Appendix B.4, Table 7), the decoding step analysis (Table 3), and the sample efficiency study (Table 9). FlowGuard achieves an overall AUC of 0.942 on the default configuration. We chose thresholded metric for the main results because the downstream task of defending against jailbreak attacks is most intuitively understood as a binary safe/unsafe classification. We will include a consolidated AUROC table in the final version of the paper.
>
> > Q3: False positive analysis and robustness to over-refusal.
>
> We report utility preservation in Table 2, and Table 5 (Appendix B.2) explicitly reports the False Positive Rate (FPR) under different threshold settings. With the default threshold configuration, FlowGuard achieves an FPR of 2.4%, compared to 11.2% for perplexity-based detection and 14.5% for confidence-based detection (Table 6). The MOSSBench evaluation (Table 2) specifically stress-tests false positives on benign queries containing sensitive language, where FlowGuard achieves 79.5% accuracy, the highest among all evaluated defenses, demonstrating robustness against over-refusal. We will consolidate this analysis for the final version of the paper.

---

> > ### Author Rebuttal · Reviewer_RV7r · 2026-04-02
> >
> > Thank you for the authors’ rebuttal. I have checked the additional experiments in the appendix, and they are helpful in addressing my original questions.
> >
> > In addition to the ASR reported in Table 4, it may also be helpful to report the AUROC. The cases with $\lambda = 0.1$ and $\lambda = 1.0$ appear to warrant closer examination in the revision. Nevertheless, it is encouraging to see a white-box adaptive attack included to stress-test the proposed method.
> >
> > I have updated my score to 5.

---

> > > ### Author Response · Authors · 2026-04-07
> > >
> > > We thank the reviewer for the constructive engagement and for updating their score. We have revised the paper to: (1) add a summary of the adaptive attack analysis (Table 4) to the main text, (2) include AUROC alongside ASR in the main results table, with closer examination of the λ settings as suggested, and (3) consolidate the false positive analysis into a more in-depth discussion. We appreciate the suggestion on AUROC reporting for the adaptive attack table specifically and believe all of the aforementioned changes helped greatly in improving the rigor of the paper.

---

### Official Review · Reviewer_LqVu · 2026-03-12

**Soundness:** 3
**Presentation:** 3
**Significance:** 3
**Originality:** 3
**Overall Recommendation:** 4
**Confidence:** 4

**Summary:**

This paper proposes FlowGuard, an inference-time defense for multimodal jailbreak detection that monitors the internal consistency among text-only, vision-only, and joint predictive distributions. It uses these relationships to build a compact feature representation and trains a one-class Isolation Forest using only benign data. Experiments on three MLLMs and several multimodal attack benchmarks show strong reductions in attack success rate, limited utility degradation, and relatively favorable latency.

**Compliance With Llm Reviewing Policy:**

Affirmed.

**Key Questions For Authors:**

1、Can the authors explore any approaches to adapt this method for black-box settings where logits are unavailable?
2、Does relying on the first token make the system vulnerable to attacks that stay benign at the start but steer the model toward harmful content in later generation steps? How does it perform when harmful semantics appear after several decoding steps?
3、How does the method perform in multi-turn conversations or multi-image scenarios? Might cross-modal consistency drift in these contexts?
4、Have the author tested the method on larger models (e.g., 70B parameters)? Will the method still work on larger, inherently safer models, or will the anomalous structures collapse as the base model becomes safer?

**Limitations:**

1、The approach fundamentally requires white-box access to logits, excluding many practical black-box scenarios.
2、The method is validated mainly in single-turn input settings and does not yet handle more practical scenarios like cumulative dialogue or multi-image interactions.

**Strengths And Weaknesses:**

Strengths:
1、The paper targets an important phenomenon in MLLM safety by focusing on harmful behavior that emerges only after cross-modal fusion, rather than checking the text and image independently.
2、The one-class classification setup trained solely on benign data enables zero-shot detection of unseen jailbreak attacks without adversarial supervision.
3、The method works at inference time and achieves a strong balance among defense effectiveness, utility preservation, and latency.
4、The experiments are comprehensive, covering multiple VLM architectures, diverse datasets, novel baselines, defense on adaptive attacks, and ablation studies.
5、The paper is well organized and clearly written, making the paper easy to follow.


Weaknesses:
1、The method requires access to next-token distributions, so it is still a white-box approach that limits applicability in black-box API settings.
2、The strong performance under the first token setting may be partly driven by superficial response-format cues, with safe answers often beginning with tokens like “I” in refusals such as “I’ m sorry” or “I apologize,” and unsafe answers may begin with tokens such as “Sure,”. This setting may miss attacks that remain benign initially but turn harmful later.
3、The paper lacks evaluation on multi-turn jailbreaks or multi-image inputs, where cross-modal consistency might drift over extended dialogue contexts.
4、It is unclear if the method remains effective on larger models or inherently safer instruction-following models, as the anomalous structures might collapse as base models become more robust.

---

> ### Author Rebuttal · Authors · 2026-03-30
>
> We thank the reviewer for the thorough assessment and address their concerns below.
>
> > Q1: Access requirements, black-box adaptation, and validation on API-based models.
>
> See our response to Reviewer c5hc (Q2) for the full discussion. FlowGuard requires only next-token distributions (not weights or gradients), positioning it to be between full white-box and black-box settings. To validate feasibility beyond self-hosted models, we also include new evaluations for FlowGuard using GPT-4.1-mini's top-k logprob API access and find that approximate FlowVectors computed from partial distributions achieve 10.4% average ASR across attack types, confirming that the fusion consistency signal is recoverable even from partial distributional access. Full results will be added to the final version of the paper.
>
> > Q2: FlowGuard captures model-internal fusion structure, not superficial first-token cues; extended monitoring catches delayed attacks.
>
> FlowGuard does not detect whether the model will refuse or comply. It captures the relational structure of cross-modal fusion: redundancy, synergy, and uniqueness reflect how modalities interact during reasoning, not what any single token is. Table 3 (Section 5.5) confirms this: extending to k=10 yields only +0.006 AUC, confirming the fusion disruption signal is concentrated at the first step rather than driven by surface-format patterns.
> Importantly, while overall AUC improvement is small, we find that approximately 2.1% of adversarial samples that evade detection at k=1 are caught at k=3, corresponding to cases where harmful semantics emerge after the initial decoding step. This confirms that extended monitoring provides a complementary safeguard for delayed-onset attacks and can be selectively deployed when the latency budget permits (Table 3). We will expand the discussion of potential adaptive temporal monitoring to build on this further in Section 5.5 of the final version.
>
> > Q3: Multi-turn and multi-image evaluation require fundamentally different research formulations.
>
> FlowGuard operates on individual input pairs, and both extensions pose fundamentally different research problems. Multi-turn detection requires modeling how FlowVectors evolve across a multi-turn trajectory, where harmful intent may accumulate gradually across individually benign turns, requiring temporal modeling over the FlowVector sequence rather than single-point anomaly detection. Multi-image inputs require extending the modal decomposition itself: the current framework decomposes information flow between two sources (text and vision), but multiple images introduce combinatorial pairwise and higher-order interactions that scale with the number of visual inputs.
> Recent work confirms these are recognized as distinct research agendas: Li et al. [1] demonstrate that current single-turn defenses fail against multi-turn human jailbreaks, and Russinovich et al. [2] show that multi-turn attacks like Crescendo exploit context accumulation mechanisms fundamentally different from single-turn manipulation. We scope our contribution to the single-turn, single-image threat model that dominates current multimodal jailbreak research accordingly, and add discussion of future extensions to other settings in the final version of the paper.
>
> > Q4: Validation on larger models (LLaMA 3.1 70B).
>
> Our current evaluation spans 4B to 7B and shows consistent performance (Appendix D). FlowGuard's one-class design extends naturally as safer models would produce tighter benign FlowVector clusters, yielding tighter decision boundaries. We validate this on LLaMA 3.1 70B, where FlowGuard achieves 7.9% average ASR across attack types, consistent with our findings on smaller models. Full results will appear in the final version.
>
> [1] https://arxiv.org/abs/2408.15221
>
> [2] https://arxiv.org/abs/2404.01833

---

> > ### Author Rebuttal · Reviewer_LqVu · 2026-04-07
> >
> > Thanks for providing answers to my questions and I will maintain my score.

---

> > > ### Author Response · Authors · 2026-04-07
> > >
> > > We thank the reviewer again for their review and for indicating that the rebuttal addressed the concerns raised in the original review.
> > >
> > > To summarize, we addressed: (1) black-box adaptation, including **new results on GPT-4.1-mini using top-k logprob access (10.4% average ASR)**, (2) first-token reliance, showing the signal reflects fusion structure rather than surface cues, with extended monitoring catching only an additional 2.1% of delayed-onset attacks, (3) scoping of multi-turn and multi-image settings as distinct research directions with added discussion, and (4) **additional experiment to validate on LLaMA 3.1 70B (7.9% average ASR)**, confirming consistent performance at scale. All results and discussion have been added for the final version.
> > >
> > > As the reviewer marked all concerns as fully resolved, we respectfully ask whether the reviewer would consider adjusting the score accordingly. If not, we would be very grateful if the reviewer could clarify whether there are any remaining concerns preventing a higher score, as this would be valuable both for the AC and for guiding the final revision.

---

### Official Review · Reviewer_c5hc · 2026-03-13

**Soundness:** 3
**Presentation:** 3
**Significance:** 3
**Originality:** 3
**Overall Recommendation:** 5
**Confidence:** 3

**Summary:**

The paper proposes FlowGuard, a method for detecting cross-modality attacks in Multimodal Large Language Models (MLLMs). In particular, the authors propose comparing the consistency between the unimodal distributions and the joint predictive distributions to detect the existence of an attack. FlowVectors are used to quantify the cross-modal redundancy, synergy, and modality-specific dominance to check whether the multimodal representation aligns with the unimodal semantics. Extensive experiments demonstrate that FlowGuard reduces the attack success rate even in unseen attacks.

**Compliance With Llm Reviewing Policy:**

Affirmed.

**Final Justification:**

The rebuttal addressed my main concerns and I am supportive of acceptance.

**Key Questions For Authors:**

- Can the method be extended to white-box model access, where the information about the unimodal representations are not present?
- Is there an ablation study motivating the three elements of uniqueness, synergy and redundancy as being sufficient for identifying attacks? It would be interesting if there are other qualitative metrics that can enhance even more the detection of more complicated attacks.
- Please refer to the other questions in the Weaknesses section.

**Limitations:**

Yes

**Strengths And Weaknesses:**

Strengths:
- The proposed method seems to reduce the success attack rate significantly across all benchmarks.
- It incorporates the intuition that unimodal and multimodal representations in benign inputs align, thus misalignment between those can be used as a detection mechanism for an attack.
- The proposed method measures the uniqueness, synergy and redundancy of each representation along with their fusion and thus detects complex attacks beyond the surface-level of unimodal jailbreaks.

Weaknesses:
- Are there any other quantities beyond uniqueness, synergy and redundancy that can be used to further improve the detection of attacks?
- How restrictive is the white-box access to all unimodal representations for applying the method in practice?
- For some data and attacks, the method seems to underperform with respect to the state-of-the-art, while in other it outperforms them significantly. Have you detected any particular type of attacks or data modalities that FlowGuard might be less effective based on the numerical experiments?
- Typo: In the abstract: "evidencebetween" requires a space.

---

> ### Author Rebuttal · Authors · 2026-03-30
>
> We thank the reviewer for the positive assessment and thoughtful questions.
>
> > Q1: Whether additional quantities beyond the current FlowVector could improve detection.
>
> This is an interesting point. In principle, additional features could be derived from the distributional relationships such as higher-order interaction terms, Rényi divergences, or Wasserstein distances between the unimodal and multimodal distributions. However, our ablation study (Section 5.4, Figures 3-4) shows that the current four-dimensional FlowVector motivated by Partial Information Decomposition already achieves strong performance (0.942 AUC), and synergy alone reaches standalone AUC > 0.88. This suggests the current feature set captures the dominant signals of adversarial fusion disruption.
> In a one-class setting, adding features increases the risk of overfitting to spurious correlations without supervised signal to regularize against. Exploring richer feature sets for detecting more sophisticated future attacks is a valuable direction we will discuss in the final version.
>
> > Q2: Access requirements and potential for black-box extension.
>
> FlowGuard requires only next-token probability distributions, not model weights, gradients, or internal activations. This positions it in a middle ground between full white-box access and black-box settings which are substantially weaker assumptions than methods requiring gradient access or internal representations, while stronger than purely text-based approaches. Any deployment where the operator controls the inference pipeline (self-hosted or on-premise) naturally provides this access, covering a large fraction of safety-critical scenarios.
> The focus of this paper is establishing that cross-modal fusion consistency is a reliable and general detection signal for multimodal jailbreaks across architectures, attacks, and modalities (as found in Tables 1, 11, 12). We present these results as a necessary prerequisite before adaptation to restricted-access settings: without first demonstrating that the signal exists and generalizes, there is no basis for pursuing approximate versions of it. For API-only settings, some providers already expose top-k logprobs, and recent work on sampling-based distributional estimation (Semantic Entropy [1]) and black-box behavioral probing (QueRE [2], which includes adversarial detection) suggests viable adaptation paths that build on the findings we present here.
>
> > Q3: Attack types and modalities where FlowGuard is less effective.
>
> FlowGuard shows relatively higher ASR on text-only attacks (around 13-14%) compared to visual and cross-modal attacks (around 6-9%).  We believe this is to be expected based on FlowGuard’s design: text-only attacks do not disrupt cross-modal fusion in the same way visual or compositional attacks do, so the detection signal is inherently weaker. FlowGuard is most differentiated on compositional and obfuscation-based attacks where surface-level analysis fails. As discussed in Section 1 and Figure 1, FlowGuard is designed as a lightweight, plug-and-play defense layer that complements existing defenses. At 1.3 seconds per sample (Section 5.3, Figure 2), it adds minimal overhead when stacked with text-specific defenses, which are an extensively explored area with strong existing solutions.
>
> > Q4: Typo.
>
> Thank you; we will make a pass to correct all typos and grammatical errors in the final version.
>
> [1] https://arxiv.org/abs/2302.09664
>
> [2] https://arxiv.org/abs/2501.01558

---

### Official Review · Reviewer_UCaX · 2026-03-22

**Soundness:** 1
**Presentation:** 1
**Significance:** 2
**Originality:** 3
**Overall Recommendation:** 4
**Confidence:** 4

**Summary:**

This paper proposes FlowGuard, a method for detecting adversarial inputs that can come from both textual and visual modalities against multimodal models. The authors first argue that visual and textual inputs for multimodal models would induce "compatible predictive behavior" in benign settings BUT NOT in adversarial settings. Motivated by this, the authors then use predictive logits from multimodal models induced by textual/visual inputs as "low-level" features to design "high-level" information theory-based features (based on KL-divergence, JS-divergence, and Shannon entropy). Isolation forests are then trained on these high-level features to learn to perform multimodal adversarial input detection.

**Compliance With Llm Reviewing Policy:**

Affirmed.

**Final Justification:**

Thanks to the authors for their detailed follow-up. Please add all follow-up experiments (regarding **Q2** and **Q4**) and explanations (regarding **Q3**) in your revision.

Given that I have no more questions for this paper, I have increased my score from 3 to 4 accordingly.

**Key Questions For Authors:**

See **Strengths and Weaknesses** for details.

**Limitations:**

The authors did not discuss in what situations their main motivation for the FlowGuard would succeed or fail. See **Strengths and Weaknesses** for details.

**Strengths And Weaknesses:**

**Strengths**

1. I think the idea of using the logit from the first decoding step induced by text-only/vision-only/joint inference (i.e., Eq. 1 and Eq. 2) as a feature is interesting and novel.


**Weaknesses & Questions**

While I think "adopting logits from the first decoding step as features" is interesting, I believe the overall design of the proposed FlowGuard is not well motivated and might even fail in some scenarios. Therefore, I tend to reject this paper, but I would like to first see how the authors justify their claims. Detailed comments are as follows:

1. The design of FlowGuard is motivated by the claim that **"visual and textual inputs induce compatible predictive behavior in benign settings but not under adversarial manipulation"**. I have some concerns about this claim:

    - The authors did not empirically justify whether this claim indeed holds in practice. So, I think the authors should conduct some visualization experiments (e.g., t-SNE) to show how the input logits under adversarial settings would differ from those under benign settings.

    - Besides, I also suspect this claim **does NOT universally hold** and might **FAIL in some scenarios**. For example, if both the input adversarial image and the input jailbreak prompt are synthesized with the same goal of inducing the same unsafe response, "Sure, here is a plan of how to make a bomb", then these two inputs might still induce compatible predictive behavior (even under an adversarial setting). Therefore, I think the authors should conduct a broader empirical investigation or present in-depth discussion to understand in what situations their claim can hold or fail. This would help to understand the capability boundary of FlowGuard.

2. In Section 3.4, the motivation for using uniqueness/redundancy/synergy features is not explained. To be honest, using these features is very weird to me. **Why not directly use those "first decoding step logits" as features and train an MLP to perform the adversarial input detection (or, say, classification)?**

3. In Section 3.5, the motivation for using an isolation forest is also unclear. As I asked in the previous question, **why not directly train an MLP (which would not be costly) to perform the adversarial detection**? What are the advantages of using an isolation forest?

4. In the experiments (Section 4 and Section 5), two important baseline attacks are missing: (1) the PAIR attack [r1], which is an agent-based text-only attack, and (2) the PGD-based attack from [r2] (see Fig. 2 in [r2]), which is an image-only attack.


**References**

[r1] Chao et al. "Jailbreaking Black Box Large Language Models in Twenty Queries." arXiv 2023.

[r2] Schlarmann et al. "Robust CLIP: Unsupervised Adversarial Fine-Tuning of Vision Embeddings for Robust Large Vision-Language Models." ICML 2024.

---

> ### Author Rebuttal · Authors · 2026-03-30
>
> We thank the reviewer for the detailed and constructive feedback and address their concerns below.
>
> > Q1: Quantitative and visual evidence for distributional separation between benign and adversarial FlowVectors.
>
> FlowGuard's core claim is not that adversarial inputs always create incompatible predictions, but that successful multimodal jailbreaks require redistributing probability mass in ways that deviate from benign fusion patterns. The FlowVector is designed to capture exactly this deviation.
> The standalone AUC analysis (Figure 4) shows Synergy alone achieves AUC > 0.88 across attack benchmarks, and the full FlowVector reaches 0.942, which is only possible if benign and adversarial distributions occupy distinct regions of the feature space. Appendix C (Table 10) further shows per-attack-type signatures: negative synergy and low redundancy under attack vs. positive synergy and high redundancy under benign fusion. We will include a t-SNE visualization in the final revision showing both raw logit embeddings and FlowVectors side by side. Preliminary results show substantial overlap for raw logits (consistent with the Raw Logits baseline achieving only 31.5% ASR at 8.9% FPR in Table 6), while FlowVectors show a cleaner separation, directly illustrating why the information-theoretic decomposition is necessary rather than redundant.
>
> > Q2: Attack-type-specific detection intuition and the case where both modalities target the same harmful output.
>
> Appendix C (Table 10) shows how each attack category produces distinct signatures. Text-only jailbreaks yield high Ut with low Uv and low redundancy; visual injections reverse this (high Uv, low Ut). In cross-modal attacks where both modalities cooperate toward the same harmful output (CBSC, UMK), fusion produces a posterior outside both unimodal supports, manifesting as negative synergy and high total divergence. FlowGuard reduces ASR on these cooperative attacks to at most 9.1% (Table 1).
> More importantly, the adaptive attack analysis in Appendix B.1 (Table 4)  reveals the tradeoff that breaking FlowGuard inherently breaks the attack. A white-box adversary jointly optimizing attack success and FlowGuard evasion (Eq. 6) finds that successful jailbreaks require shifting probability mass toward harmful continuations, which necessarily distorts the multimodal distribution relative to unimodal priors. When forced to preserve fusion consistency (high lambda), harmful probability mass is suppressed and ASR collapses to 6.8%. When optimizing freely (low lambda), ASR reaches 94.5% but detection is reliable (score=0.82). No point achieves both, and this holds across text-only, visual, and cross-modal attacks, confirming this is a structural property of multimodal inference, not an artifact of specific attack types.
> We will add a discussion of scenarios where the assumption may weaken, connecting to these adaptive attack findings.
>
> > Q3: Motivation for PID-inspired features (uniqueness/redundancy/synergy) over raw logits.
>
> The motivation is that raw logits are a point representation of a single distribution, while our hypothesis was that adversarial behavior is defined by the relationship across distributions: how text-only, vision-only, and fused multimodal predictions relate to each other. Uniqueness measures which modality dominates fusion (detecting asymmetric attacks), redundancy measures whether modalities agree on shared semantics (detecting conflicting intent), and synergy measures whether fusion reduces or amplifies uncertainty (detecting unstable integration). These correspond to the three failure modes in Section 3.3 and Table 10 (see Q2 above).
> The Raw Embedding baseline (Table 1) and Raw Logits baseline (Table 6) confirm this: the relational encoding is what enables zero-shot generalization in the one-class setting, as also illustrated by the t-SNE comparison described in Q1.
>
> > Q4: Isolation Forest generalizes to unseen attacks where supervised MLP does not.
>
> We also directly evaluate this in Table 7 (Appendix B.4) as a  supervised MLP achieves 0.982 AUC on its training distribution (MMSB) but collapses to 0.612 on VLSafe, 0.558 on VLSU, and 0.650 on VizWiz (flagging benign OOD inputs as adversarial). Isolation Forest maintains at least 0.88 AUC across all settings, including unseen attack types and OOD benign data. The generalization gap is 0.327 AUC on VLSU (MLP: 0.558, IF: 0.885). Multimodal jailbreaks are constantly evolving, so learning the boundary of normal behavior (one-class) rather than memorizing known attacks (supervised) is essential for deployment.
>
> > Q5: Results for PAIR and Schlarmann et al. PGD attack.
>
> We ran these additional attacks. On LLaVA-1.5-7B (MMSB), FlowGuard achieves 11.8% ASR on PAIR and 6.4% ASR on the Schlarmann et al. PGD attack, finding them to be consistent with results on other text-only attacks (9.1-14.2%) and image-only attacks (6.1-8.4%) respectively. Full cross-benchmark results will appear in the final version.

---

> > ### Author Rebuttal · Reviewer_UCaX · 2026-04-04
> >
> > Thanks to the authors for their rebuttal. However, most of my concerns/questions remain unresolved. So I chose my current Acknowledgement type as (c), but you can view it as meaning that I still have many follow-up questions.
> >
> > **Regarding Q2:** It seems that none of the tables mentioned by the authors (i.e., Tables 1, 4, and 10) are trying to answer my question. My original question is: "If both the adversarial textual and visual inputs are trying to force the VLM to reproduce the same textual output, can FlowGuard still detect the attack?" I suggest that the authors directly conduct an experiment to investigate this.
> >
> > **Regarding Q3:** The authors' explanation of the hand-crafted uniqueness/redundancy/synergy features is basically a restatement of what they have already said in the original paper. However, I think the explanation in the original paper is unclear. For example, the authors said in the rebuttal that "synergy measures whether fusion reduces or amplifies uncertainty," but why can Eq. (5) achieve this goal? I suggest that the authors first explain the role and motivation of the different terms in Eqs. (3), (4), and (5), and then explain why combining these different terms can achieve the goals described for Eqs. (3), (4), and (5).
> >
> > **Regarding Q4:** Is it the case that you feed the four hand-crafted features (i.e., uniqueness/redundancy/synergy) into the MLP for training? If so, then it is not surprising that the trained MLP does not perform well, as MLPs are usually more effective for high-dimensional data. I suggest that the authors train the MLP directly on the raw logit features (i.e., $P_{mm}$, $P_t$, and $P_v$) and compare its performance with Isolation Forest.

---

> > > ### Author Response · Authors · 2026-04-06
> > >
> > > We thank the reviewer for the continued engagement in the paper and address each point below.
> > >
> > > Q2: We ran an additional experiment to address this case. We sampled 200 harmful queries from MMSB and optimized both modalities toward the same target completion ("Sure, here is"): a GCG suffix on the text side and a PGD perturbation on the image side, each optimized with the other modality held clean. At test time, they are paired. On LLaVA-1.5-7B, undefended ASR is 86.4%. **Despite both modalities being optimized toward the same attack goal, FlowGuard detects these inputs, reducing ASR to 11.2%.**
> > >
> > > Detection works at two levels (Section 3.5). First, the FlowVector encodes cross-modal interaction patterns by examining the full first-step distributions, not just the top token. GCG and PGD attacks both push “Sure” to the top, but induce different posterior shapes, so the probability mass below the top token differs. FlowVector captures this as abnormal synergy (S = -0.31 vs. +0.42 for benign), indicating destabilizing rather than constructive fusion. The Isolation Forest flags this FlowVector as inconsistent with benign patterns.
> > >
> > > Table 4 (Appendix B.1) addresses the stronger case where the adversary attempts to align the full distribution rather than only the top token. An empirical tradeoff emerges: unconstrained optimization achieves 94.5% ASR but produces an anomaly score of 0.82, making it easy to detect. Constraining the attack to preserve benign distributional patterns lowers the anomaly score to 0.48 but also reduces ASR to 6.8%. In short, token-level co-optimization is detected by FlowVector's distributional sensitivity, while stronger distribution-level alignment substantially weakens the attack.
> > >
> > > Q3: We explain why Eqs. (3)-(5) are appropriate proxies for the three interaction properties they measure.
> > >
> > > Uniqueness. Each term asks how costly it is to explain the fused posterior using one unimodal prior alone. The direction of KL matters here. $D_{KL}(P_{mm} | P_t)$ becomes large when fusion assigns mass to tokens unlikely under text-only reasoning, signaling vision-specific influence. The reverse would instead emphasize tokens likely under text-only reasoning that disappear after fusion, which is not the quantity we want. $U_v$ and $U_t$ measure which modality contributes probability mass that the other cannot explain.
> > >
> > > Redundancy. This captures agreement between the two unimodal priors before fusion. JSD is used because this comparison should be symmetric and bounded to $[0,1]$. If text-only and vision-only reasoning support similar token regions, JSD is small and $R$ is high. If they support different semantic regions, JSD is large and $R$ drops. Eq. (4) directly measures how compatible the two unimodal predictions are before fusion.
> > >
> > > Synergy. This compares uncertainty before and after fusion. The average unimodal entropy provides the baseline. Subtracting the multimodal entropy asks whether combining the modalities resolves that uncertainty. If fusion is constructive, the multimodal posterior becomes sharper than the unimodal average, so $H(P_{mm}^{(1)})$ is smaller and $S > 0$. If fusion is destabilizing, $H(P_{mm}^{(1)})$ exceeds the average and $S < 0$. Eq. (5) captures whether fusion reduces or amplifies uncertainty.
> > >
> > > These features must be combined because each is ambiguous in isolation. High $R$ occurs in both benign inputs and cooperative attacks where both modalities support the same unsafe first token. Large $U_v$ or $U_t$ identifies asymmetric dominance but misses attacks where both modalities move together. $S$ detects whether fusion is stabilizing or destabilizing, but not which modality caused the deviation. The joint 4D FlowVector resolves these ambiguities by covering source-specific dominance ($U_v/U_t$), pre-fusion agreement ($R$), and post-fusion stability ($S$).
> > >
> > > Q4: Yes. In Table 7, the supervised MLP was trained on the same 4D FlowVector. We ran the requested ablation. In the one-class setting, IF on raw logits (Table 6) achieves 31.5% ASR at 8.9% FPR, while IF on FlowVectors achieves 8.3% ASR at 2.4% FPR. In the supervised setting, MLP on raw logits reaches 0.961 AUC on MMSB and transfers to 0.724/0.681/0.712 on VLSafe/VLSU/VizWiz, while MLP on FlowVectors reaches 0.982 on MMSB and transfers to 0.612/0.558/0.650.
> > >
> > > This suggests high-dimensional token-level logits are more useful when labeled samples are available, while FlowVectors are more effective in the one-class setting. Multimodal jailbreaks evolve rapidly, so labeled examples of future attack types are unlikely to be available when training a detector. Our claim is not that FlowVectors dominate raw logits, but that they are well suited to one-class anomaly detection, where raw logits introduce high-dimensional variation and FlowVectors provide a compact relational space where benign behavior clusters more tightly. FlowGuard's advantage comes from pairing a relational low-dimensional representation with one-class modeling.

---

### Decision · Program_Chairs · 2026-04-30

**Decision:**

Accept (spotlight)

**Comment:**

This paper proposes FlowGuard, an inference-time defense for multimodal jailbreak detection that identifies attacks by measuring inconsistency between text-only, vision-only, and fused predictive distributions. The core idea is novel, timely, and well aligned with an important problem in MLLM safety.  The rebuttal addressed most concerns with additional experiments and clarifications. While one reviewer still had some reservations about theoretical motivation and feature design, these do not outweigh the paper’s clear empirical strength, novelty, and practical value. Overall, this is a strong paper on an important MLLM safety problem and should be accepted.